# PHRASE GROUNDING-BASED STYLE TRANSFER FOR SINGLE-DOMAIN GENERALIZED OBJECT DETECTION

## ABSTRACT

This paper focuses on a more challenging scenario of single-domain generalized object detection, which aims to learn a detector that performs well on multiple unseen target domains with only one source domain for training. Recently, the grounded language-image pre-training model (GLIP) has gained widespread attention, which reformulates object detection as a phrase grounding task by aligning each region or box to phrases in a textual prompt. Inspired by this, this paper proposes a phrase grounding-based style transfer (PGST) approach for single-domain generalized object detection. Specifically, we introduce a textual prompt that contains a set of phrases for each target domain, such as *a car driving in the foggy scene*. Subsequently, we use the corresponding target textual prompt to train the PGST module from the source domain to the target domain, and the training losses include the localization loss and region-phrase alignment loss from GLIP. As such, the visual features of the source domain could be close to imaginary counterparts in the target domain while preserving their semantic content. When freezing PGST, we fine-tune the image and text encoders of GLIP using the style-transferred visual features of the source domain, to enhance the generalization of the model to corresponding unseen target domains. Our proposed approach significantly outperforms existing state-of-the-art methods, achieving a mean average precision (mAP) improvement of 8.9% on average across five diverse weather driving benchmarks. In addition, our performance on some datasets surprisingly matches or even surpasses that of those domain adaptive object detection methods, even though these methods incorporate target domain images into their training process.

## 1 INTRODUCTION

With the development of deep learning (Krizhevsky et al., 2012; Simonyan & Zisserman, 2015; He et al., 2016; Huang et al., 2017; Liu et al., 2021c), there have been breakthrough advancements in the object detection task within computer vision (Ren et al., 2017; He et al., 2020; Tian et al., 2019). These object detection models typically exhibit excellent performance but often rely on the assumption that the training and test datasets follow the same distribution to ensure their effectiveness. However, in open environments, the variability in the test dataset distribution due to factors such as environments, devices, and human intervention necessitates the need to annotate a large amount of data to fit any potential data distribution that may arise. This requires a significant amount of manual and computational resources. To this end, domain adaptation techniques have gained widespread attention in recent years. Their objective is to enhance the generalization of a model trained on a training set (source domain) to a test set (target domain), where these two domains exhibit some degree of relevance but follow different distributions (Zhang, 2019; Jiang et al., 2022b). Recently, domain adaptation techniques have found extensive applications in computer vision tasks such as image classification (Liu et al., 2021a; Wang et al., 2022; 2023b), semantic segmentation (Cheng et al., 2021; 2023; Wang et al., 2023c), and object detection (Chen et al., 2018; Saito et al., 2019; Zhu et al., 2019), delivering outstanding performance.

However, domain adaptation typically requires data from both source and target domains to participate in the training process, severely limiting the model's ability to generalize to an unseen target domain. Therefore, researchers have started to explore domain generalization in recent years (Wang et al., 2023a; Zhou et al., 2023), and aim to improve the generalization of a model trained on one or more source domains to unseen target domains. Furthermore, existing domain generalization techniques

often rely on multiple annotated source domains to obtain a highly generalized model, which undoubtedly incurs substantial annotation costs. Therefore, single-domain generalization (SDG), as a more challenging research topic, seeks to enhance the generalization of a model trained on a single source domain to multiple related target domains, making it a more valuable area of research.

While there have been some methods proposed to address the problem of SDG in image classification (Fan et al., 2021; Qiao et al., 2020; Volpi et al., 2018; Wang et al., 2021b; Zhang et al., 2022), research on SDG in the context of object detection remains relatively under-explored. This is partly due to the increased complexity of object detection compared to image classification. Wu & Deng (2022) are the first to deal with SDG for object detection, and they present a method of cyclic-disentangled self-distillation, to disentangle domain-invariant features from domain-specific features without the supervision of domain-related annotations. Taking advantage of contrastive language-image pre-training (CLIP) model (Radford et al., 2021), Vidit et al. (2023) design a semantic augmentation method to transform visual features from the source domain into the style of the target domain. They then construct a text-based classification loss supervised by the text embeddings from CLIP to train the model. However, CLIP typically provides visual representations at the image level and does not capture visual representations at the object level for the object detection task. This limitation results in the style transfer method being less accurate, as it can only perform global style transfer and not at the object level. Recently, Li et al. present a grounded language-image pre-training (GLIP) model, which reformulates object detection as a phrase grounding task and aims to learn an object-level, language-aware, and semantic-rich visual representations.

Building upon the advantages of GLIP in the object detection task, we propose a novel phrase grounding-based style transfer (PGST) approach for single-domain generalized object detection. Specifically, we first define a textual prompt that utilizes a set of phrases to describe potential objects for each unseen target domain, such as *persons or cars in road scenes with different weather conditions*. Then, we use the corresponding target textual prompt to train the proposed PGST module from the source domain to the target domain and adopt the training losses of the localization loss and region-phrase alignment loss from GLIP. As a result, the visual features of the source domain could be close to imaginary counterparts in the target domain while preserving their semantic content. Finally, we fine-tune the image and text encoders of GLIP using these style-transferred source visual features. As such, the detector in GLIP could be well generalized to unseen target domains using only one single source domain for training. Our main contributions can be summarized below:

1) To the best of our knowledge, we are the first to apply the GLIP model to a more challenging and practical domain adaptation scenario, namely single-domain generalized object detection, which has not been thoroughly explored to date. 2) To leverage the advantages of the GLIP model, we introduce a phrase-based style transfer approach by pre-defining textual prompts, which aims to enable the visual features from the source domain to be close to imaginary counterparts in the target domain. As a result, the fine-tuned GLIP model with a style-transferred source domain could obtain desirable performance on corresponding unseen target domains. 3) We evaluate the performance of our proposed approach on five different weather driving benchmarks, achieving a mean average precision (mAP) improvement of 8.9% on average across five diverse weather driving benchmarks. Moreover, our proposed approach on some datasets surprisingly matches or even surpasses domain adaptive object detection methods, which incorporate target domain images into the training process.

## 2 RELATED WORK

### 2.1 DOMAIN ADAPTIVE OBJECT DETECTION

Domain adaptive object detection (DAOD) aims to enhance the generalization of an object detector trained on a well-labeled source domain to an unlabeled target domain. It assumes that both labeled data from the source domain and unlabeled data from the target domain could be involved in the training process. Existing DAOD methods can be broadly divided into three categories: feature learning-based methods, data manipulation-based methods, and learning strategy-based methods. Among these, feature learning-based methods employ strategies such as adversarial learning Chen et al. (2018); Saito et al. (2019); Zhu et al. (2019), class prototype alignment (Xu et al., 2020b; Zheng et al., 2020; Zhang et al., 2021; Chen et al., 2021), and feature disentanglement (Su et al., 2020; Wu et al., 2021; 2022; Wu & Deng, 2022) to learn domain-invariant features between the source and target domains. These strategies can be applied at different feature extraction stages of the object

detection model. Data manipulation-based methods directly augment (Kim et al., 2019b; Khirodkar et al., 2019; Prakash et al., 2019; Wang et al., 2021a) or perform style transformations (Huang et al., 2018; Rodriguez & Mikolajczyk, 2019; Yun et al., 2021; Yu et al., 2022; Hsu et al., 2020) on input data to narrow the distribution gap between the source and target domains. Learning strategy-based methods achieve object detection of target domain by introducing some learning strategies like self-training (RoyChowdhury et al., 2019; Zhao et al., 2020a; Li et al., 2021) and teacher-student networks (Cai et al., 2019; He et al., 2022; Li et al., 2022b). However, DAOD assumes that target domain data is visible during the training phase, which limits its practical applicability in real-life scenarios. In contrast, the single-domain generalized object detection task addressed in this paper aims to generalize the model trained on one source domain to multiple unseen target domains, which is a more challenging and practical scenario.

## 2.2 SINGLE-DOMAIN GENERALIZATION

SDG aims to improve the performance on unseen target domains with only one source domain for training, and the limited diversity may jeopardize the model's generalization on unseen target domains. To tackle this problem, Volpi et al. (2018) propose an iterative procedure that augments the dataset with examples from a fictitious target domain that is *hard* under the current model. To facilitate fast and desirable domain augmentation, Qiao et al. (2020) cast the model training in a meta-learning scheme and use a Wasserstein auto-encoder to relax the widely used worst-case constraint. Wang et al. (2021b) propose a style-complement module to enhance the generalization power of the model by synthesizing images from diverse distributions that are complementary to the source ones. Zhang et al. (2022) cast domain generalization as a feature distribution matching problem, and propose to perform exact feature distribution matching by exactly matching the empirical cumulative distribution functions of image features. Based on the adversarial domain augmentation, Fan et al. (2021) propose a generic normalization approach, where the statistics are learned to be adaptive to the data coming from different domains, and hence improve the model's generalization performance across domains. While SDG has been well-studied in the image classification task, it has not received widespread attention in the object detection task. This is because object detection task involves both classification and localization, making them a relatively less-explored area. Wu & Deng (2022) are the first to deal with single-domain generalized object detection and present a method of cyclic-disentangled self-distillation. With the vision-language model CLIP (Radford et al., 2021), Vidit et al. (2023) design a semantic augmentation method to transform visual features from the source domain into the style of the target domain. In contrast, the proposed model is based on GLIP (Li et al., 2022a) which could learn object-level, language-aware, and semantic-rich visual representations.

## 2.3 VISION-LANGUAGE MODELS

Recently, developing vision-language models to address computer vision tasks has become a prominent trend. For instance, CLIP (Radford et al., 2021) and ALIGN (Jia et al., 2021) perform cross-modal contrastive learning on hundreds or thousands of millions of image-text pairs, enabling direct open-vocabulary image classification task. By distilling the knowledge from CLIP/ALIGN into a two-stage detector, ViLD (Gu et al., 2021) is introduced to tackle the zero-shot object detection task. Alternatively, MDETR (Kamath et al., 2021) trains an end-to-end model on existing multi-modal datasets that have explicit alignment between phrases in text and objects in an image. GLIP (Li et al., 2022a) reformulates object detection as a phrase grounding task, to obtain a good grounding model and simantic-rich representations, simultaneously. To the best of our knowledge, we are the first to work on addressing the single-domain generalized object detection using the GLIP model.

## 3 PROPOSED METHOD

The transferability of pre-trained models is a new paradigm in the era of deep learning for solving cross-domain tasks (Chen et al., 2023). This paper explores how to use the pre-trained model, *i.e.*, GLIP (Li et al., 2022a), to address the single-domain generalized object detection task that has still not been fully explored (Wu & Deng, 2022). GLIP jointly trains an image encoder and a text encoder on 27M image-text pairs, enhancing performance in both object detection and phrase grounding tasks. Furthermore, GLIP could learn object-level, language-aware, and semantic-rich visual representations through region-phrase alignment loss, effectively bridging the two different modalities. In this paper,

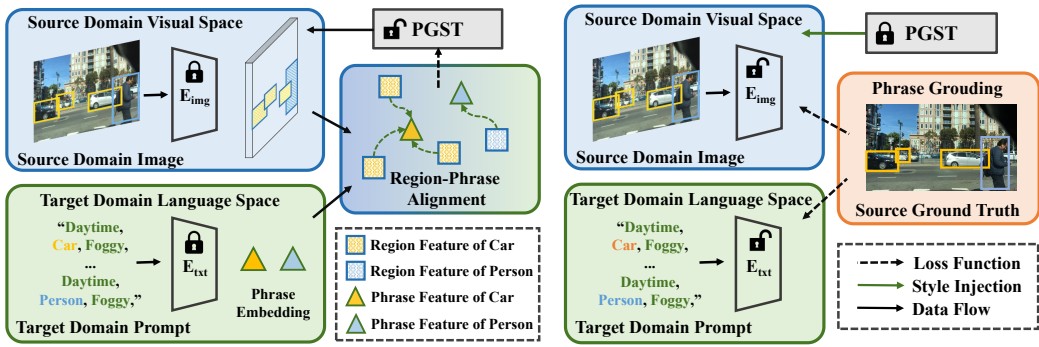

Figure 1: Phrase grounding-based style transfer for single-domain generalized object detection.

we leverage these advantages of GLIP and design a textual prompt that contains a set of phrases to describe each target domain well, such as *a car driving in the daytime and foggy scene*. We achieve style transfer from visual features of the source domain to the target domain while preserving their semantic content within the GLIP embedding space using these prompts. Our goal is to fine-tune the model by using these style-transferred visual features of the source domain, thereby enhancing the model's generalization during the inference process in the unseen target domains.

**Problem Formulation.** In our task, there exists a single labeled source domain denoted as $\mathcal{D}_s = \{(\mathbf{X}_s^i, \mathbf{B}_s^i, \mathbf{Y}_s^i)\}_{i=1}^{n_s}$, and any unlabeled target domain denoted as $\mathcal{D}_t = \{\mathbf{X}_t^i\}_{i=1}^{n_t}$, where the source domain and target domain follow different distributions. Here, $\mathbf{X}^i$ represents an image, $\mathbf{B}^i$ represents the bounding box coordinates, and $\mathbf{Y}^i$ represents the categories. We only employ the source domain to train an object detector that can achieve satisfactory performance on multiple unseen target domains. The objectives of this paper can be mainly divided into two aspects: 1) Training a style transfer module for each target domain, achieving style transfer from the source domain to the target domain; 2) Under the supervision of the style transfer module, fine-tuning the object detector $\mathbf{G}_s^{\text{det}}$ in GLIP using the source domain. This detector consists of an image encoder $\mathbf{E}_{\text{img}}$ and a text encoder $\mathbf{E}_{\text{txt}}$, aiming to achieve generalization of the detector on unseen target domains.

**Overview of PGST.** To achieve the aforementioned objectives, this paper introduces a novel phrase grounding-based style transfer (PGST) approach for single-domain generalized object detection, primarily consisting of three crucial stages. As shown in Figure 1(a), after freezing the image and text encoders within GLIP, we compute the localization loss and region-phrase alignment loss by inputting images of the source domain and pre-defined prompt of the target domain. We use these losses to train the proposed PGST module from the source domain to the target domain. As depicted in Figure 1(b), we freeze the PGST module and perform style transfer on the visual features from the source domain to the target domain. Then, we fine-tune the image encoder and text encoder of GLIP to enhance the model's generalization to corresponding unseen target

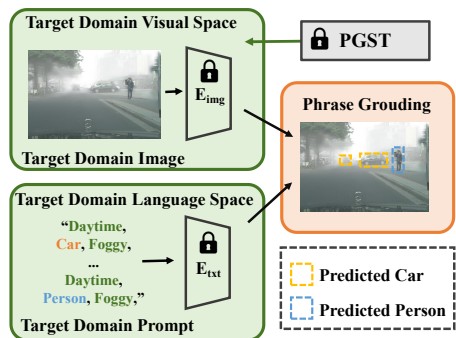

Figure 2: Inference on unseen domain.

domains. Finally, as shown in Figure 2, we input images from the unseen target domain along with the corresponding prompt into the model to complete the inference process for the unseen target domain. The primary challenge is how to achieve PGST from the source domain to the target domain for the object detection task, while preserving their semantic content, even in the absence of target domain images. We will provide a detailed explanation in the following section on how to define prompts for the unseen target domains and design the PGST module.

## 3.1 PHRASE GROUNDING-BASED STYLE TRANSFER

To achieve better style alignment, we initially develop prompts for both the source and target domains. These prompts serve to strengthen the semantic connection between visual and language features,

thereby enabling the training of a more robust source-only detection model. Additionally, we create a style transfer module to enhance the detection model's generalization capability.

**Textual Prompt Definition.** We develop two prompt templates including the domain-specific prompt and the general prompt. As for the specific-domain prompt, we construct a textual prompt using a series of phrases. For example, when dealing with a driving dataset captured in foggy conditions, we incorporate prefixes or suffixes around category labels to generate phrases that describe the category within the specific domain. These phrases are then concatenated with commas to form the textual prompt for the target domain. In order to improve the source-only object detection model, we define a textual prompt in the source domain. In the general prompt, we include all potential stylistic information from the unseen target domains while adding prefixes and suffixes. For instance, we might have phrases like "a car driving in the sunny, foggy, or rainy scene." The domain-specific prompt aims to facilitate style transfer from the source domain to a given target domain where the images are not accessible, thereby improving the model's domain-specific generalization. In contrast, the general prompt is employed to enhance the model's generalization on a series of potential unseen target domains. For more details, please refer to Appendix C.

**Phrase Grounding-Based Style Transfer.** Designing an effective style transfer method is crucial for improving the generalization of a source domain-trained model to unseen target domains. Adaptive instance normalization (AdAIN) (Huang & Belongie, 2017), as a simple and effective style transfer technique, utilizes the mean and standard deviation to characterize the style of a domain. It then stylizes a source domain feature, to match the style of the target domain with the supervision of these two statistical measures. Fahes et al. (2023) utilize AdAIN for style transfer in CLIP, to deal with domain adaptation in the semantic segmentation task. Furthermore, there are notable differences in their optimization strategies for AdAIN and prompt design compared to our proposed approach.

Inspired by this, PGST injects the style parameters of the target domain, which include the aforementioned mean and standard deviation, into the low-level features of the source domain extracted from the backbone network, to achieve a style transformation of the source domain. Let us denote the low-level feature of the source domain as $\mathbf{f}_s \in \mathbb{R}^{h \times w \times c}$, where $h$, $w$, and $c$ represent the height, width, and number of channels, respectively. Then, the style of the source domain can be abstracted as $\boldsymbol{\mu}_s = \mu(\mathbf{f}_s)$ and $\boldsymbol{\sigma}_s = \sigma(\mathbf{f}_s)$. Here, $\mu(\cdot)$ and $\sigma(\cdot)$ are two functions to respectively return the channel-wise mean and standard deviation of the features. Therefore, stylizing source domain feature $\mathbf{f}_s$ with any target domain style $(\boldsymbol{\mu}_t, \boldsymbol{\sigma}_t)$ can be formalized as the following equation,

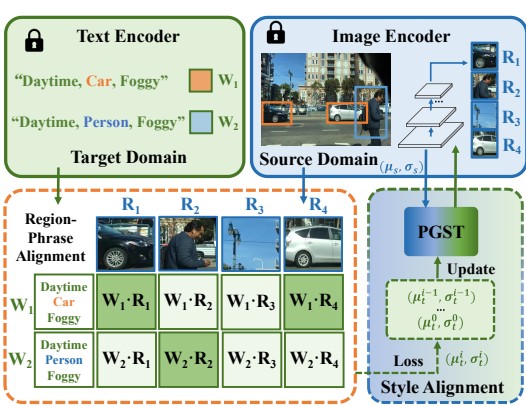

Figure 3: Phrase grounding-based style transfer.

$$\text{PGST}_{s \to t}(\mathbf{f}_s, \boldsymbol{\sigma}_t, \boldsymbol{\mu}_t) = \boldsymbol{\sigma}_t \left( \frac{\mathbf{f}_s - \boldsymbol{\mu}_s}{\boldsymbol{\sigma}_s} \right) + \boldsymbol{\mu}_t, \tag{1}$$

where $\boldsymbol{\mu}_t$ and $\boldsymbol{\sigma}_t$ represent the channel-wise mean and standard deviation of the target domain features, or target domain style. However, traditional style transfer methods typically require access to target domain images $\mathbf{f}_t$, and perform a global style transfer across the entire image. Therefore, for the single-domain generalized object detection task addressed in this paper, we face two major challenges: 1) How to learn the style of the target domain $\boldsymbol{\mu}_t$ and $\boldsymbol{\sigma}_t$ without having seen target domain images, enabling the style transfer from the source domain to the target domain. 2) How to achieve style transfer at the object level to better suit the object detection task.

To address the aforementioned challenges, this paper treats the target domain style $\boldsymbol{\mu}_t$ and $\boldsymbol{\sigma}_t$ as variables to be optimized. Through the definition of target domain prompt and the utilization of the region-phrase alignment loss in GLIP, it accomplishes object-level style learning and transfer. As shown in Figure 3, we initialize $\boldsymbol{\mu}_t$ and $\boldsymbol{\sigma}_t$ using the style parameter of low-level feature $\mathbf{f}_s$ in the source domain. Next, we employ the image encoder and text encoder in GLIP to extract the object regions of interest for the source domain image and obtain phrase embeddings from the target domain prompt, respectively. To learn the style of the target domain and achieve style transfer at an object

level, we minimize the following loss function,

$$\mathbf{R} = \mathrm{E}_{\mathrm{img}}(\mathrm{Img}_s), \quad \mathbf{W} = \mathrm{E}_{\mathrm{txt}}(\mathrm{Prompt}_t), \quad \mathcal{L}_{\mathrm{Ground}} = \mathbf{R}\mathbf{W}^{\top}, \tag{2}$$

$$\mathcal{L}_{\boldsymbol{\mu}_t, \boldsymbol{\sigma}_t}(\bar{\mathbf{f}}_s, \mathbf{b}_s, \mathbf{y}_s) = \mathcal{L}_{\mathrm{Loc}} + \mathcal{L}_{\mathrm{Ground}}, \tag{3}$$

where $\mathbf{R}_{i=1}^{\mathrm{Nr}} \in \mathbb{R}^{h \times w \times d}$ denotes object regions whose number is Nr, and $\mathbf{W}_{i=1}^{\mathrm{Nw}} \in \mathbb{R}^{1 \times d}$ denotes phrase embeddings whose number is Nw. $\mathcal{L}_{\mathrm{Loc}}$ represents the localization loss in a two-stage detector of Faster R-CNN (Ren et al., 2017). $\bar{\mathbf{f}}_s$ is the style transferred visual feature of source domain. $\mathcal{L}_{\mathrm{Ground}}$ denotes the region-phrase alignment loss following (Li et al., 2022a), aiming to align object regions of source domain and phrases of the target domain prompt. Since the visual features of the source domain need to align with the phrases in the target domain prompt at an object level, $\boldsymbol{\mu}_t$ and $\boldsymbol{\sigma}_t$ gradually approach the true style of the target domain as the loss from Equation 3 back-propagated, resulting in a more accurate style transfer from the source domain to the target domain. Notably, since high-level features contain more semantic information, in order to preserve semantic information during the style transfer process, we only perform style transfer on low-level features. Furthermore, the learning process of this style transfer is conducted at an object level rather than an image level, which also ensures that the semantic information can be preserved. Specifically, we obtain $\boldsymbol{\mu}_t$ and $\boldsymbol{\sigma}_t$ for each source domain image, forming a collection of style transfer parameters $\mathcal{T}_{s \to t}$. This ensures that we can adequately describe the style information of the target domain.

## 3.2 FINE-TUNING FOR GENERALIZATION

Following the training criteria proposed by (Li et al., 2022a), we utilize source domain images and target domain prompts as inputs for training PGST. Then, we randomly sample a set of style transfer parameters from $\mathcal{T}_{s \to t}$ and apply style transfer to the low-level visual features of the source domain, generating corresponding object regions from the image encoder. Besides, we obtain target domain phrase embeddings with the text encoder. Finally, we fine-tune both the image encoder and text encoder based on the loss functions within the GLIP model. Since this paper considers generalization to multiple unseen target domains, we define a prompt and train style transfer modules for each target domain. Subsequently, we fine-tune GLIP for each target domain separately and evaluate the model's generalization performance on the corresponding target domain images.

## 4 EXPERIMENTS

The foundational detection model used in our study is the pre-trained GLIP-T[1], which consists of the Swin-Tiny (Liu et al., 2021c) backbone and Dynamic detection head (Dai et al., 2021) for image encoding, as well as the BERT encoder (Devlin et al., 2018) for text encoding. All models are trained on 4 RTX3090 GPUs with 24G RAM, and our code will be open-sourced very soon.

## 4.1 IMPLEMENTATION DETAILS

**Datasets.** We evaluate the proposed model using the same dataset as (Wu & Deng, 2022), which consists of images from five different weather conditions in urban scenes: daytime sunny, night sunny, dusk rainy, night rainy, and daytime foggy. Specifically, we select images from the Berkeley Deep Drive 100k (BDD-100k) dataset (Yu et al., 2020) for the daytime sunny scene. This subset includes 19,395 training images for model training and 8,313 test images for validation and model selection. Images from the night sunny scene are also chosen from the BDD-100k dataset, which contains 26,158 images. Additionally, images from the dusk rainy and night rainy scenes are obtained from (Wu et al., 2021), which consists of 3,501 and 2,494 images, respectively. Finally, images from the daytime foggy scene are collected from the Foggy Cityscapes (Cordts et al., 2016) and Adverse Weather (Hassaballah et al., 2020) datasets, amounting to a total of 3,775 images. All datasets are annotated with seven object categories and corresponding bounding boxes, including *bus*, *bicycle*, *car*, *motor*, *person*, *rider*, and *truck*. Examples from the five scenes are illustrated in Figure 4. Our proposed model is trained solely on the training set of the daytime sunny scene (source domain) and is tested on the other four challenging weather conditions (unseen target domains).

---

[1] https://github.com/microsoft/GLIP

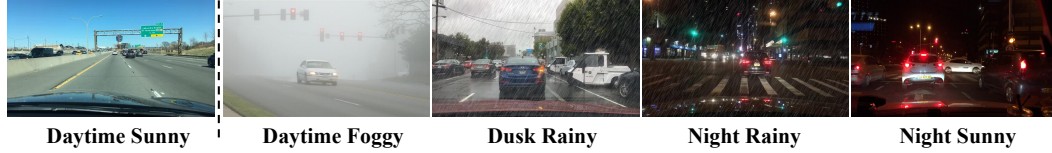

| Daytime Sunny | Daytime Foggy | Dusk Rainy | Night Rainy | Night Sunny |

Figure 4: Exemplary images of five different weather conditions.

**Metrics.** We evaluate the performance of the proposed model using the mean Average Precision (mAP) metric with the Intersection-over-Union (IoU) threshold of 0.5 following (Wu & Deng, 2022).

**Source Domain Augmentation with Prompt.** We define a textual prompt in the source domain and employ a full-model tuning strategy for source domain augmentation following (Li et al., 2022a). This not only results in a good source-only model but also greatly aids in subsequent style transfer. More specifically, we use the AdamW (Loshchilov & Hutter, 2019) optimizer with a learning rate of 0.0001 and a weight decay of 0.05. We train our source-only model for 12 epochs with a batch size of 8. To save training costs, we resize the input images such that the longest side is no more than 1,166 pixels, while the shortest side is 480 pixels.

**Phrase Grounding-Based Style Transfer.** When training the proposed PGST module, we freeze the weights of the source-only model, which has been enhanced with the prompt in the source domain. Moreover, we use the stochastic gradient descent optimizer with a learning rate of 1.0, a momentum of 0.9, and a weight decay of 0.0001. The prompt that we design for the target domain follows the format of *Time*, *category*, in the *weather* scene. To enrich the semantic information of specific objects, such as *rider*, we introduce phrases like *person who rides a bicycle or motorcycle*. It is worth noting that we develop two prompt templates: the general prompt and the domain-specific prompt, and we use the former as the default prompt template for reporting evaluation results. For the complete prompt definitions related to this paper, please refer to Appendix C.

**Training for Generalization.** To enhance the model's generalization, we employ a labeled source domain, textual prompt of the target domain, and a well-trained PGST module to fine-tune the model. This training process is similar to the source domain augmentation, with the exception that we apply the PGST module to low-level features of the backbone network for style transfer. Additionally, we employ the training-domain validation set strategy for model selection following C-Gap (Vidit et al., 2023; Gulrajani & Lopez-Paz, 2020).

## 4.2 COMPARISON WITH THE STATE OF THE ART

We compare our proposed model with two existing single-domain generalized object detection approaches: S-DGOD (Wu & Deng, 2022) and C-Gap (Vidit et al., 2023). Following (Wu & Deng, 2022), we also compare several feature normalization generalization methods, including SW (Pan et al., 2019), IBN-Net (Pan et al., 2018), IterNorm (Huang et al., 2019), and ISW (Choi et al., 2021).

From Table 1, it can be observed that the model proposed in this paper achieves significant improvements over the best baseline method, C-Gap (Vidit et al., 2023), in all four unseen target domains, with gains of 11.0%, 12.2%, 9.7%, and 4.0%, respectively. Moreover, even on the test set of the source domain itself, the proposed model outperforms the best baseline method, S-DGOD (Wu & Deng, 2022), by 11.0% (Table 2). Additionally, we illustrate the object detection performance on four different unseen target domains in Figure 5. These results collectively demonstrate the effectiveness of the model proposed in this paper for the single-domain generalized object detection problem.

Table 1: Single domain generalization (mAP).

| Method | Daytime Sunny | Night Sunny | Dusk Rainy | Night Rainy | Daytime Foggy |
|---|---|---|---|---|---|
| F-RCNN (2017) | 51.1 | 33.5 | 26.6 | 14.5 | 31.9 |
| SW (2019) | 50.6 | 33.4 | 26.3 | 13.7 | 30.8 |
| IBN-Net (2018) | 49.7 | 32.1 | 26.1 | 14.3 | 29.6 |
| IterNorm (2019) | 43.9 | 29.6 | 22.8 | 12.6 | 28.4 |
| ISW (2021) | 51.3 | 33.2 | 25.9 | 14.1 | 31.8 |
| S-DGOD (2022) | 56.1 | 36.6 | 28.2 | 16.6 | 33.5 |
| C-Gap (2023) | 51.3 | 36.9 | 32.3 | 18.7 | 38.5 |
| **Ours** | **63.7** (+7.6) | **47.9** (+11.0) | **44.5** (+12.2) | **28.4** (+9.7) | **42.5** (+4.0) |

Table 2: Per-class results (%) on Daytime Sunny. Our approach outperforms S-DGOD by 9.7%.

| Method | Bus | Bike | Car | Mot. | Pers. | Rider | Truck | mAP |
|---|---|---|---|---|---|---|---|---|
| F-RCNN (2017) | 63.4 | 42.9 | 53.4 | 49.4 | 39.8 | 48.1 | 60.8 | 51.1 |
| SW (2019) | 62.3 | 42.9 | 53.3 | 49.9 | 39.2 | 46.2 | 60.6 | 50.6 |
| IBN-Net (2018) | 63.6 | 40.7 | 53.2 | 45.9 | 38.6 | 45.3 | 60.7 | 49.7 |
| IterNorm (2019) | 58.4 | 34.2 | 42.4 | 44.1 | 31.6 | 40.8 | 55.5 | 43.9 |
| ISW (2021) | 62.9 | 44.6 | 53.5 | 49.2 | 39.9 | 48.3 | 60.9 | 51.3 |
| S-DGOD (2022) | **68.8** | 50.9 | 53.9 | **56.2** | 41.8 | 52.4 | **68.7** | 56.1 |
| C-Gap (2023) | 53.6 | 46.7 | 66.4 | 44.8 | 48.0 | 45.7 | 53.8 | 51.6 |
| **Ours** | 64.8 | **57.3** | **79.7** | 51.4 | **66.8** | **58.6** | 67.2 | **63.7** |

| **Daytime Foggy** | **Dusk Rainy** | **Night Rainy** | **Night Sunny** |
|---|---|---|---|

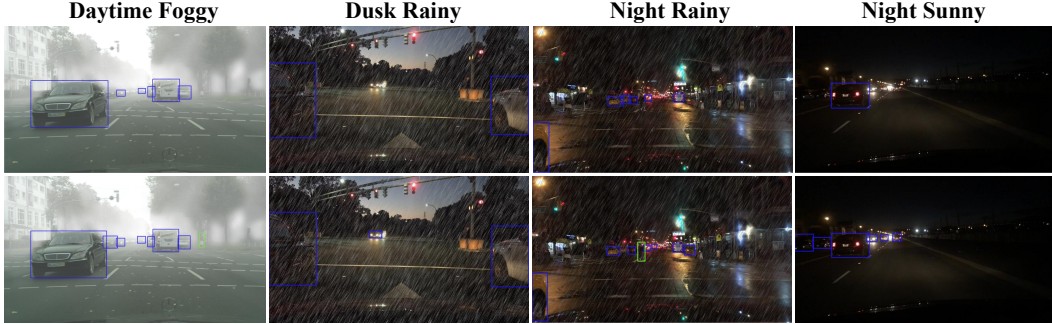

Figure 5: Visualization results of object detection. (Top): GLIP-T. (Bottom): the proposed model. Upon comparison, it can be observed that the proposed model is capable of detecting two distinct objects, the person (marked by green boxes) and the car (marked by blue boxes), more accurately than the baseline model GLIP-T.

Tables 3 to 6 report the detection results for each category in different domains. Below, we analyze the results for the four unseen target domains.

Table 3: Per-class results (%) on Daytime Foggy.

| Method | Bus | Bike | Car | Mot. | Pers. | Rider | Truck | mAP |
|---|---|---|---|---|---|---|---|---|
| F-RCNN (2017) | 28.1 | 29.7 | 49.7 | 26.3 | 33.2 | 35.5 | 21.5 | 32.0 |
| S-DGOD (2022) | 32.9 | 28.0 | 48.8 | 29.8 | 32.5 | 38.2 | 24.1 | 33.5 |
| C-Gap (2023) | 36.1 | 34.3 | 58.0 | 33.1 | 39.0 | **43.9** | 25.1 | 38.5 |
| **Ours** | **41.7** | **36.4** | **65.4** | **35.9** | **44.9** | 42.8 | **30.2** | **42.5** |

Table 4: Per-class results (%) on Dusk Rainy.

| Method | Bus | Bike | Car | Mot. | Pers. | Rider | Truck | mAP |
|---|---|---|---|---|---|---|---|---|
| F-RCNN (2017) | 28.5 | 20.3 | 58.2 | 6.5 | 23.4 | 11.3 | 33.9 | 26.0 |
| S-DGOD (2022) | 37.1 | 19.6 | 50.9 | 13.4 | 19.7 | 16.3 | 40.7 | 28.2 |
| C-Gap (2023) | 37.8 | 22.8 | 60.7 | 16.8 | 26.8 | 18.7 | 42.4 | 32.3 |
| **Ours** | **50.9** | **33.8** | **73.6** | **23.9** | **46.0** | **27.1** | **56.9** | **44.6** |

Table 5: Per-class results (%) on Night Sunny.

| Method | Bus | Bike | Car | Mot. | Pers. | Rider | Truck | mAP |
|---|---|---|---|---|---|---|---|---|
| F-RCNN (2017) | 34.7 | 32.0 | 56.6 | 13.6 | 37.4 | 27.6 | 38.6 | 34.4 |
| S-DGOD (2022) | 40.6 | 35.1 | 50.7 | 19.7 | 34.7 | 32.1 | 43.4 | 36.6 |
| C-Gap (2023) | 37.7 | 34.3 | 58.0 | 19.2 | 37.6 | 28.5 | 42.9 | 36.9 |
| **Ours** | **50.3** | **49.0** | **69.5** | 17.5 | **56.0** | **39.3** | **53.4** | **47.9** |

Table 6: Per-class results (%) on Night Rainy.

| Method | Bus | Bike | Car | Mot. | Pers. | Rider | Truck | mAP |
|---|---|---|---|---|---|---|---|---|
| F-RCNN (2017) | 16.8 | 6.9 | 26.3 | 0.6 | 11.6 | 9.4 | 15.4 | 12.4 |
| S-DGOD (2022) | 24.4 | 11.6 | 29.5 | 9.8 | 10.5 | 11.4 | 19.2 | 16.6 |
| C-Gap (2023) | 28.6 | 12.1 | 36.1 | 9.2 | 12.3 | 9.6 | 22.9 | 18.7 |
| **Ours** | **42.9** | **19.5** | **49.8** | 3.5 | **27.5** | **18.6** | **37.1** | **28.4** |

**Daytime Sunny to Daytime Foggy.** In contrast to images taken in clear, sunny conditions, those captured under foggy weather display varying degrees of blurriness. As illustrated in Table 3, our proposed model outperforms the recent state-of-the-art methods, S-DGOD (Wu & Deng, 2022) and C-Gap (Vidit et al., 2023), across 6 categories, underlining its robust generalization capabilities.

**Daytime Sunny to Dusk Rainy.** The dim light and interference from rain during dusk rainy conditions present significant challenges that can compromise a model's generalization. As illustrated in Table 4, our proposed model outshines the recent state-of-the-art methods, S-DGOD (Wu & Deng, 2022) and C-Gap (Vidit et al., 2023), across all categories. Notably, there's a marked improvement in the *person* category, emphasizing its robust generalization capabilities.

**Daytime Sunny to Night Sunny.** Object detection under the veil of nighttime darkness is inherently challenging, especially for categories such as *person*, *rider*, and *motor*, which are prone to misidentification. As demonstrated in Table 5, our proposed model outperforms the two leading methods, S-DGOD (Wu & Deng, 2022) and C-Gap (Vidit et al., 2023), across 6 categories. This underscores the robust generalization capabilities of our model.

**Daytime Sunny to Night Rainy.** Night rainy conditions, compounded by the dual challenges of darkness and rain, present arguably the toughest generalization task among the four unexplored target domains. As demonstrated in Table 6, our proposed model outshines the two prominent state-of-the-art methods, S-DGOD (Wu & Deng, 2022) and C-Gap (Vidit et al., 2023), clinching the top performance in 6 out of the 7 categories, even under these severe conditions. It is noteworthy, however, that the exceptionally adverse weather conditions often blur the distinctions between the *bike*, *rider*, and *motor* categories, leading to subpar results for these three. Nonetheless, the resilience and generalization prowess of our model are palpably affirmed in this challenging scenario.

### 4.3 EMPIRICAL ANALYSIS

To evaluate the individual contributions of the various modules in our model, we first omit the source domain augmentation with prompt and the PGST, then assess the model's generalization across diverse target domains (referring to Table 7). During the training, we immobilize the model's image encoder to ascertain the indispensability of the source domain transfer (see Table 10). Additionally, we examine the influence of unrelated prompts (see Table 14) and domain-specific prompts (see Table 12) on the sensitivity of our model. Moreover, to discern the efficacy of PGST at varying feature extraction depths, we implement it across different layers of the backbone, particularly focusing on its impact at low-level features (see Table 11).

In the main text, it is mentioned that we initially train the model of GLIP-T using a labeled source domain along with a corresponding prompt. As the source domain provides accurate and rich labeling information, this allows the model to learn common concepts across domains. The four rows in Table 7 represent the performances of pre-trained GLIP-T, source domain augmentation, PGST, and the proposed full model, respectively. It can be observed that our proposed model with prompt augmentation for the source domain results in a significant improvement in performance, both within its own domain and in other unseen target domains, compared to using pre-trained GLIP-T.

However, due to substantial distribution differences between the source domain and unseen target domains, the model's performance in unseen target domains is much lower than within the source domain itself. Therefore, there is still considerable room for improvement in the model's generalization performance. As indicated in the third and fourth rows of Table 7, when we utilize the target domain prompt to learn the target domain style and realize style transfer for the source domain, there is a substantial performance boost in the four unseen target domains compared to pre-trained GLIP-T (pre-trained GLIP-T with source domain augmentation), with improvements of 20.0% (4.1%), 19.2% (4.4%), 17.6% (4.4%), and 15.2% (5.1%), respectively. These results highlight the significant role played by the style transfer module (PGST) proposed in this paper in enhancing the generalization of the GLIP-T model.

Table 7: Ablation study.

| Model | | Source | Target | | | | |
|---|---|---|---|---|---|---|---|
| Src. Aug. | PGST & Tuning. | Day Sunny | Night Sunny | Dusk Rainy | Night Rainy | Day Foggy |
| | | 31.6 | 24.9 | 24.1 | 12.1 | 26.5 |
| ✓ | | **63.7** | 43.8 | 40.1 | 24.0 | 37.4 |
| | ✓ | 31.6 | 45.1 | 43.1 | 25.0 | 40.9 |
| ✓ | ✓ | **63.7** | **47.9** | **44.5** | **28.4** | **42.5** |

For a comprehensive review and analysis, kindly refer to Appendix A to D.

### 5 CONCLUSION AND DISCUSSION

In this paper, we have proposed a novel phrase grounding-based style transfer approach for the single-domain generalized object detection task. This task is not only highly practical but also presents a significant challenge, yet it remains relatively unexplored. Specifically, we leverage the advantages of the GLIP model in the object detection task. By defining prompts for unseen target domains, we learn the style of these target domains and achieve style transfer from the source domain to the target domain. This style transfer process benefits from the region-phrase alignment loss employed in GLIP, ensuring a more accurate transfer. Subsequently, we employ the style-transferred visual features from the source domain, along with their corresponding annotations, to fine-tune the GLIP model. This leads to improved performance (generalization) on unseen target domains. Extensive experimental results have demonstrated that the proposed approach can achieve state-of-the-art results.

However, we only employ a combination of weather and time to formulate the textual prompt for the domain. Although empirical experiments have demonstrated the feasibility of this concise design, it has not fully harnessed the advantages of phrase grounding. In future endeavors, we can individually craft more nuanced phrases for each category within the domain and apply our proposed PGST to a vision-language model with richer semantic concepts (Yao et al., 2022; 2023), thereby further enhancing the model's generalization capability.

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

# A    COMPARISON RESULTS

To further validate the effectiveness of our proposed model in the single-domain generalized object detection problem, we conduct additional experiments to assess its generalization on other mainstream cross-domain object detection datasets. Specifically, we utilize PASCAL VOC 2007 dataset (Everingham et al., 2010) (real scenes) as the single source domain and Clipart (Inoue et al., 2018) and Watercolor (Inoue et al., 2018) as unseen target domains (cartoon-style). Due to significant stylistic differences between the source and target domains, transferring the object detector trained on the source domain to the unseen target domains poses a challenging task.

To this end, we design a simple textual prompt for the two unseen target domains by adding a suffix. For the Clipart dataset, the phrase format is designed as *class, clipart*. For the Watercolor dataset, the phrase format is designed as *class, watercolor*. Additionally, we maintain the same experimental settings as the other experiments mentioned in the main text.

From Tables 8 and 9, it can be observed that the proposed model achieves results similar to or even higher than those of the domain adaptive object detection (DAOD) approaches, which, however, assume access to target domain images during training. One key factor contributing to this is that we leverage the advantages of the GLIP model in object detection. Another significant factor is that our designed style transfer module, *i.e.*, PGST with domain-specific prompt, further greatly enhances the generalization of the GLIP model.

Table 8: Results from PASCAL VOC 2007 to Clipart. The results of DAOD methods are from (Jiang et al., 2022a).

| Method | aero | bcycle | bird | boat | bottle | bus | car | cat | chair | cow | table | dog | hrs | bike | prsn | plnt | sheep | sofa | train | tv | mAP |
|---|---|---|---|---|---|---|---|---|---|---|---|---|---|---|---|---|---|---|---|---|---|
| F-RCNN (2017) | 35.6 | 52.5 | 24.3 | 23.0 | 20.0 | 43.9 | 32.8 | 10.7 | 30.6 | 11.7 | 13.8 | 6.0 | 36.8 | 45.9 | 48.7 | 41.9 | 16.5 | 7.3 | 22.9 | 32.0 | 27.8 |
| DA-Faster (2018) | 15.0 | 34.6 | 12.4 | 11.9 | 19.8 | 21.1 | 23.2 | 3.1 | 22.1 | 26.3 | 10.6 | 10.0 | 19.6 | 39.4 | 34.6 | 29.3 | 1.0 | 17.1 | 19.7 | 24.8 | 19.8 |
| BDC-Faster (2019) | 20.2 | 46.4 | 20.4 | 19.3 | 18.7 | 41.3 | 26.5 | 6.4 | 33.2 | 11.7 | 26.0 | 1.7 | 36.6 | 41.5 | 37.7 | 44.5 | 10.6 | 20.4 | 33.3 | 15.5 | 25.6 |
| WST-BSR (2019a) | 28.0 | 64.5 | 23.9 | 19.0 | 21.9 | 64.3 | 43.5 | 16.4 | 42.0 | 25.9 | 30.5 | 7.9 | 25.5 | 67.6 | 54.5 | 36.4 | 10.3 | 31.2 | 57.4 | 43.5 | 35.7 |
| SWDA (2019) | 26.2 | 48.5 | 32.6 | 33.7 | 38.5 | 54.3 | 37.1 | 18.6 | 34.8 | 58.3 | 17.0 | 12.5 | 33.8 | 65.5 | 61.6 | 52.0 | 9.3 | 24.9 | 54.1 | 49.1 | 38.1 |
| MAF (2019) | 38.1 | 61.1 | 25.8 | 43.9 | 40.3 | 41.6 | 40.3 | 9.2 | 37.1 | 48.4 | 24.2 | 13.4 | 36.4 | 52.7 | 57.0 | 52.5 | 18.2 | 24.3 | 32.9 | 39.3 | 36.8 |
| CRDA (2020a) | 28.7 | 55.3 | 31.8 | 26.0 | 40.1 | 63.6 | 36.6 | 9.4 | 38.7 | 49.3 | 17.6 | 14.1 | 33.3 | 74.3 | 61.3 | 46.3 | 22.3 | 24.3 | 49.1 | 44.3 | 38.3 |
| Unbiased (2021b) | 30.9 | 51.8 | 27.2 | 28.0 | 31.4 | 59.0 | 34.2 | 10.0 | 35.1 | 19.6 | 15.8 | 9.3 | 41.6 | 54.4 | 52.6 | 40.3 | 22.7 | 28.8 | 37.8 | 41.4 | 33.6 |
| D-adapt (2022a) | 56.4 | 63.2 | 42.3 | 40.9 | 45.3 | 77.0 | 48.7 | 25.4 | 44.3 | 58.4 | 31.4 | 24.5 | 47.1 | 75.3 | 69.3 | 43.5 | 27.9 | 34.1 | 60.7 | 64.0 | 49.0 |
| GLIP (2022a) | 61.4 | 74.6 | 50.1 | 29.9 | 60.7 | 63.4 | 50.7 | 20.0 | 68.3 | 44.6 | 42.1 | 24.4 | 40.3 | 82.3 | 77.9 | 68.4 | 3.4 | 45.6 | 50.7 | 56.5 | 50.8 |
| **Ours** | 62.1 | 79.1 | 47.7 | 41.2 | 70.3 | 86.5 | 58.8 | 25.9 | 74.3 | 41.6 | 52.1 | 24.2 | 39.3 | 91.3 | 80.7 | 78.3 | 19.8 | 39.5 | 49.7 | 63.2 | 56.3 |

Table 9: Results from PASCAL VOC 2007 to WaterColor. The results of DAOD methods are from (Jiang et al., 2022a).

| Method | bike | bird | car | cat | dog | prsn | mAP |
|---|---|---|---|---|---|---|---|
| F-RCNN (2017) | 68.8 | 46.8 | 37.2 | 32.7 | 21.3 | 60.7 | 44.6 |
| BDC-Faster (2019) | 68.6 | 48.3 | 47.2 | 26.5 | 21.7 | 60.5 | 45.5 |
| DA-Faster (2018) | 75.2 | 40.6 | 48.0 | 31.5 | 20.6 | 60.0 | 46.0 |
| WST-BSR (2019a) | 75.6 | 45.8 | 49.3 | 34.1 | 30.3 | 64.1 | 49.9 |
| MAF (2019) | 73.4 | 55.7 | 46.4 | 36.8 | 28.9 | 60.8 | 50.3 |
| SWDA (2019) | 82.3 | 55.9 | 46.5 | 32.7 | 35.5 | 66.7 | 53.3 |
| MCAR (2020b) | 87.9 | 52.1 | 51.8 | 41.6 | 33.8 | 68.8 | 56.0 |
| UMT* (2021) | 88.2 | 55.3 | 51.7 | 39.8 | 43.6 | 69.9 | 58.1 |
| D-adapt (2022a) | 77.4 | 54.0 | 52.8 | 43.9 | 48.1 | 68.9 | 57.5 |
| GLIP (2022a) | 77.6 | 51.6 | 54.3 | 35.5 | 29.9 | 72.8 | 53.6 |
| **Ours** | 85.6 | 52.0 | 53.8 | 41.6 | 36.3 | 74.1 | 57.2 |

# B    EMPIRICAL ANALYSIS

**Feature Visualization.** To gain further insights into the reasons behind the good performance of our proposed model, we employed the t-SNE feature visualization algorithm (Van der Maaten & Hinton, 2008) to analyze the feature distributions before and after using the style transfer module (PGST). As shown in Figure 6, it visualizes the transformation of visual features from the source domain to four different unseen target domains. For example, in Figure 6(a), we transfer features from **daytime sunny** (purple) style to **daytime foggy** (claybank), and it can be observed that the transferred **daytime sunny** features (green) and **daytime foggy** (claydark) style features are closely

aligned. Furthermore, similar visual effects can be observed in the other three sub-figures. These results collectively demonstrate the effectiveness of the proposed style transfer module, which plays a crucial role in enhancing the generalization of the GLIP model.

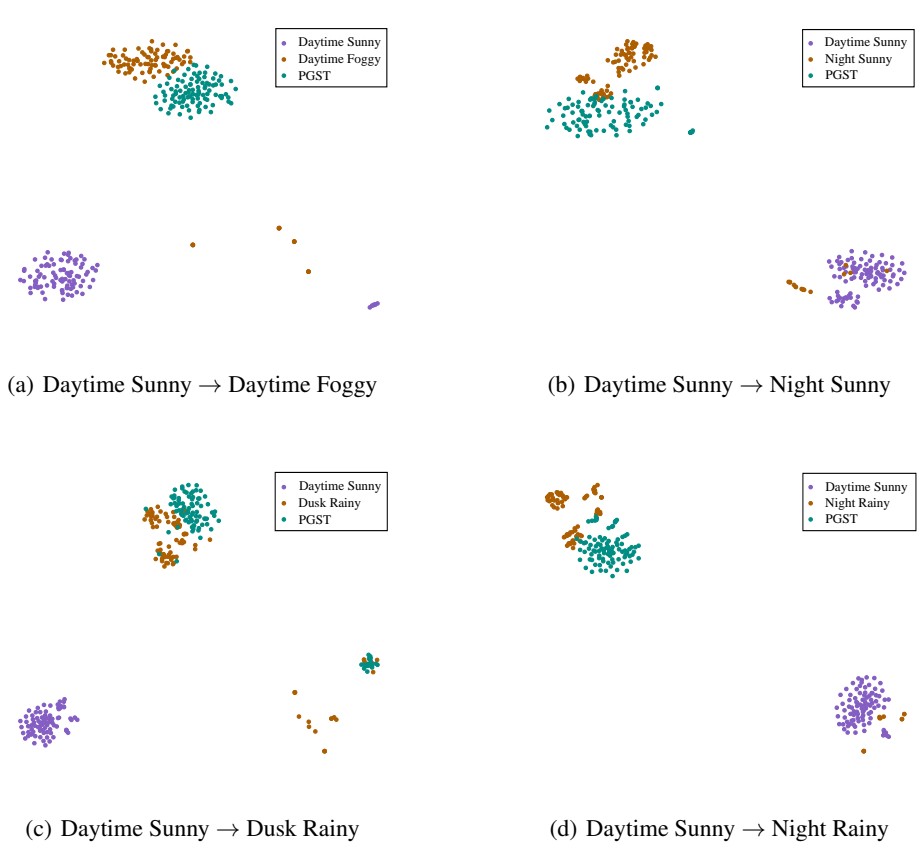

(a) Daytime Sunny → Daytime Foggy

(b) Daytime Sunny → Night Sunny

(c) Daytime Sunny → Dusk Rainy

(d) Daytime Sunny → Night Rainy

Figure 6: t-SNE visualization of low-level features.

**Number of Iterations for PGST.** In our PGST, we perform 100 optimization iterations on the source image features to achieve style alignment. The impact of the total number of iterations on the unseen target domains is illustrated in Figure 7. We observe a rapid increase in mAP from 0 to 100 iterations, followed by a gradual improvement, reaching mAP of 43.2 (Daytime Foggy) and 29.7 (Night Rainy). Moreover, there are noticeable inflection points occurring between 400-500 iterations. In support of this observation, we turn to (Kwon & Ye, 2022) to provide evidence for the presence of the "over-stylization" issue in this particular context.

**Prompt Tuning.** While fine-tuning GLIP-T with source domain augmentation and PGST, we update both the image and text encoders concurrently (full-model tuning). In Table 10, we assess the model's generalization capability with a frozen image encoder (prompt tuning), juxtaposing the results with those in Table 7. It is evident the overall efficacy delineated in Table 10 lags behind that of Table 7. This can be attributed to the pre-trained GLIP-T model en-

Table 10: mAP results when training the model with prompt tuning strategy.

| Model | | Source | Target | | | | |
|---|---|---|---|---|---|---|---|
| Src. Aug. | PGST & Tuning. | Day Sunny | Night Sunny | Dusk Rainy | Night Rainy | Day Foggy | |
| | | 31.6 | 24.9 | 24.1 | 12.1 | 26.5 | |
| ✓ | | **59.4** | 32.2 | 32.1 | 16.1 | 32.1 | |
| ✓ | ✓ | **59.4** | **40.0** | **39.4** | **22.0** | **39.8** | |

capsulating visual concepts resonant with the target domain, rendering a degree of transferability (Li et al., 2022a). Additionally, our hypothesis posits the visual semantics inherent to the source domain hold more relevant compared to the target domain. This not only yields superior results but also emphasizes the indispensable nature of transferability.

**PGST for Different Layers.** The proposed style transfer module, PGST, in our study predominantly operates on the first layer's features (low-level) of the backbone. To understand its adaptability, we evaluate its performance on other layers-specifically the 3rd and 5th layers—as delineated in Table 11. Notably, the optimal performance is witnessed when the style transfer occurs on the first layer. We posit that the more advanced layers

Table 11: PGST with a domain-specific prompt for different layers.

| Layer-1 | Layer-3 | Layer-5 | Day Foggy | Night Rainy |
|---------|---------|---------|-----------|-------------|
| ✓ | | | **42.7** | **27.2** |
| ✓ | | ✓ | 41.4 | 26.0 |
| ✓ | ✓ | ✓ | 40.3 | 25.1 |

carry richer semantic details. Executing style transfer on these elevated layers might integrate the target domain's style, potentially overshadowing the source domain's semantic essence, which in turn could impinge upon the model's efficiency.

**General Prompt vs. Domain-Specific Prompt.** In our default experimental setting, we utilize a general prompt to train PGST, ensuring that we are in the domain generalization setting and maintaining fairness in comparison with C-Gap (Vidit et al., 2023). Moreover, we incorporate a domain-specific prompt tailored to the given target domain, to enhance the model's generalization on a specific scenario and validate the robustness of our approach to prompt design. As reported in Table 12, we observe that for the Dusk Rainy and Night Rainy domains, employing PGST with a domain-specific prompt yields superior performance. In contrast, for the remaining two domains, the general prompt is proven to be more effective in terms of generalization performance. However, the results of both prompt templates are nearly identical, underscoring the robustness of our proposed approach to prompt design.

Table 12: Comparison results with different prompt templates.

| Method | Night Sunny | Dusk Rainy | Night Rainy | Daytime Foggy |
|--------|-------------|------------|-------------|---------------|
| PGST (w/ general prompt) | **47.9** | 44.5 | **28.4** | 42.5 |
| PGST (w/ domain-specific prompt) | 46.6 | **45.1** | 27.2 | **42.7** |

**Comparison Fairness.** GLIP has its own object detector and we are the first to apply GLIP to single domain generalized object detection problem. Moreover, we adopt the region-phrase alignment loss in the object detector to optimize the style transfer module. Therefore, it is non-trivial to achieve a fair comparison you mentioned. To emphasize the crucial role of the proposed design in performance improvements, we conduct experiments and record the results in Table 13, including Faster-RCNN, C-Gap (based on the large-scale pre-trained model CLIP and employing Faster-RCNN) (Vidit et al., 2023), GLIP, and our proposed design. It can be observed that, on one hand, our proposed method exhibits a significant improvement over GLIP. On the other hand, the improvement of our proposed design over GLIP is more substantial compared to the improvement of C-Gap over Faster-RCNN. These experimental results indicate the effectiveness of our proposed design.

Table 13: Comparison results with different models.

| Method | Night Sunny | Dusk Rainy | Night Rainy | Daytime Foggy |
|--------|-------------|------------|-------------|---------------|
| Faster RCNN (2017) | 33.5 | 26.6 | 14.5 | 31.9 |
| C-Gap (2023) | 36.9 (10.1%) | 32.3 (21.4%) | 18.7 (29.0%) | 38.5 (20.7%) |
| GLIP (2022a) | 32.2 | 32.1 | 16.1 | 32.1 |
| PGST (Ours) | **45.1 (40.1%)** | **43.1 (34.3%)** | **25.0 (55.3%)** | **40.8 (27.4%)** |

**Unrelated Prompts.** We also evaluate the proposed model using prompts that are unrelated to weather, as shown in Table 14. To generate prompts unrelated to weather, we employ ChatGPT to generate prefixes and suffixes for each category in unseen target domains. In Table 14, we illustrate the effects of two different prompt configurations: **A** includes the prefix "dreamlike setting" and the suffix "in the scene exudes a boring atmosphere" and **B** includes the prefix "ineffective" and the suffix "crawl slowly through the desert". We observe that the performance will be degraded. As these weather-unrelated prompts make it difficult to learn the essential differences between domains, the effectiveness of style transfer is diminished.

Table 14: Analysis for prompts that are unrelated to weather.

| Method | Night Sunny | Dusk Rainy | Night Rainy | Daytime Foggy |
|--------|-------------|------------|-------------|---------------|
| PGST (w/ domain-specific prompt) | **46.6** | **45.1** | **27.2** | **42.7** |
| PGST (w/ **A**) | 44.0 | 40.5 | 24.2 | 37.8 |
| PGST (w/ **B**) | 43.2 | 40.7 | 24.6 | 37.2 |

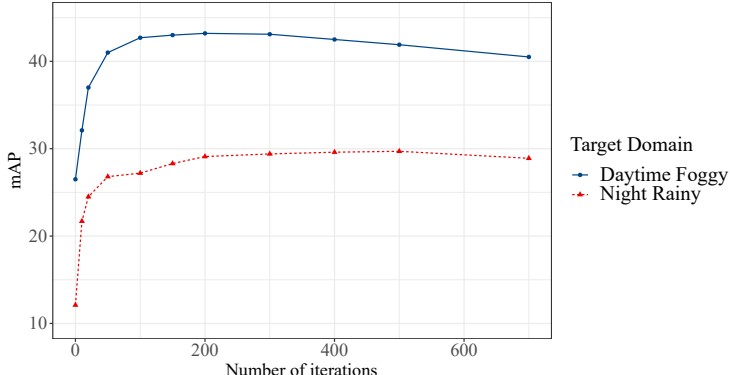

Figure 7: Effect of the number of iterations of PGST on generalization performance for tow unseen target domains.

## C  PROMPT DESIGN

The general prompt design are shown in Table 15. The prompts defined for the datasets corresponding to five different weather conditions are shown in Tables 16 to 20.

Table 15: General prompts for target domains.

| |
|---|
| daytime, dusk, night, bus, foggy, sunny, rainy, |
| daytime, dusk, night, bike, foggy, sunny, rainy, |
| daytime, dusk, night, car, foggy, sunny, rainy, |
| daytime, dusk, night, motor, foggy, sunny, rainy, |
| daytime, dusk, night, person, foggy, sunny, rainy, |
| daytime, dusk, night, rider, foggy, sunny, rainy, |
| daytime, dusk, night, truck, foggy, sunny, rainy |

Table 16: Prompt for Daytime Sunny.

| |
|---|
| daytime, bus, in the clear scene, |
| daytime, bike, in the clear scene, |
| daytime, car, in the clear scene, |
| daytime, motor, in the clear scene, |
| daytime, person, in the clear scene, |
| daytime, rider, in the clear scene, |
| daytime, truck, in the clear scene |

Table 17: Prompt for Daytime Foggy.

| |
|---|
| daytime, bus, foggy, |
| daytime, bike, bicycle, foggy, |
| daytime, car, foggy, |
| daytime, motor, motorcycle, foggy, |
| daytime, person, foggy, |
| daytime, rider, person who rides a bicycle or motorcycle in the foggy scene, |
| daytime, truck, foggy |

Table 18: Prompt for Dusk Rainy.

| |
|---|
| dusk, bus, rainy, |
| dusk, bike, bicycle, rainy, |
| dusk, car, rainy, |
| dusk, motor, motorcycle, rainy, |
| dusk, person, rainy, |
| dusk, rider, person who rides a bicycle or motorcycle in the rainy scene, |
| dusk, truck, rainy |

Table 19: Prompt for Night Rainy.

| |
|---|
| night, bus, rainy, |
| night, bike, bicycle, rainy, |
| night, car, rainy, |
| night, motor, motorcycle, rainy, |
| night, person, rainy, |
| night, rider, person who rides a bicycle or motorcycle in the rainy scene, |
| night, truck, rainy |

Table 20: Prompt for Night Sunny.

| |
|---|
| night, bus, sunny, |
| night, bike, bicycle, sunny, |
| night, car, sunny, |
| night, motor, motorcycle, sunny, |
| night, person, sunny, |
| night, rider, person who rides a bicycle or motorcycle in the sunny scene, |
| night, truck, sunny |

## D  INFERENCE VISUALIZATION RESULTS

Figures 8 to 13 display additional inference visualization results of the proposed model on unseen target domains. The visualization uses consistent color boxes to represent the same objects, and only displays the box which class score higher than 0.5.

Daytime Foggy

Figure 8: Visualization results of object detection on Daytime Sunny→Daytime Foggy.

Dusk Rainy

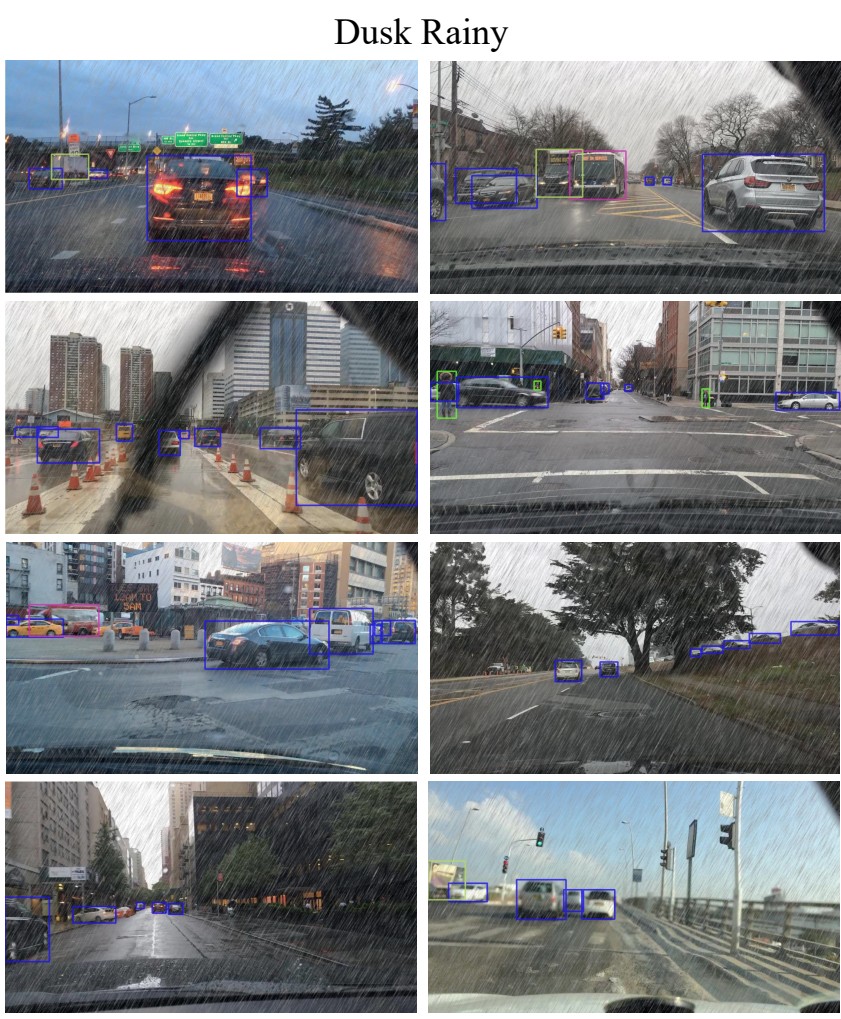

Figure 9: Visualization results of object detection on Daytime Sunny→Dusk Rainy.

Night Rainy

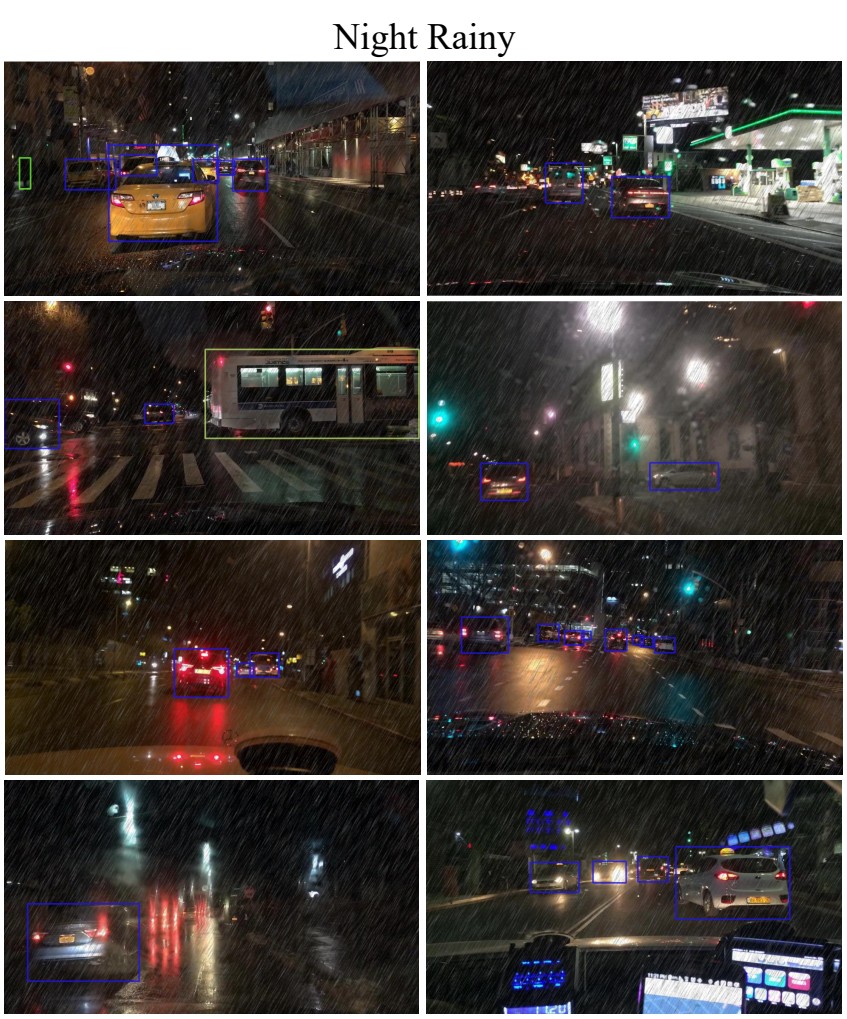

Figure 10: Visualization results of object detection on Daytime Sunny→Night Rainy.

Night Sunny

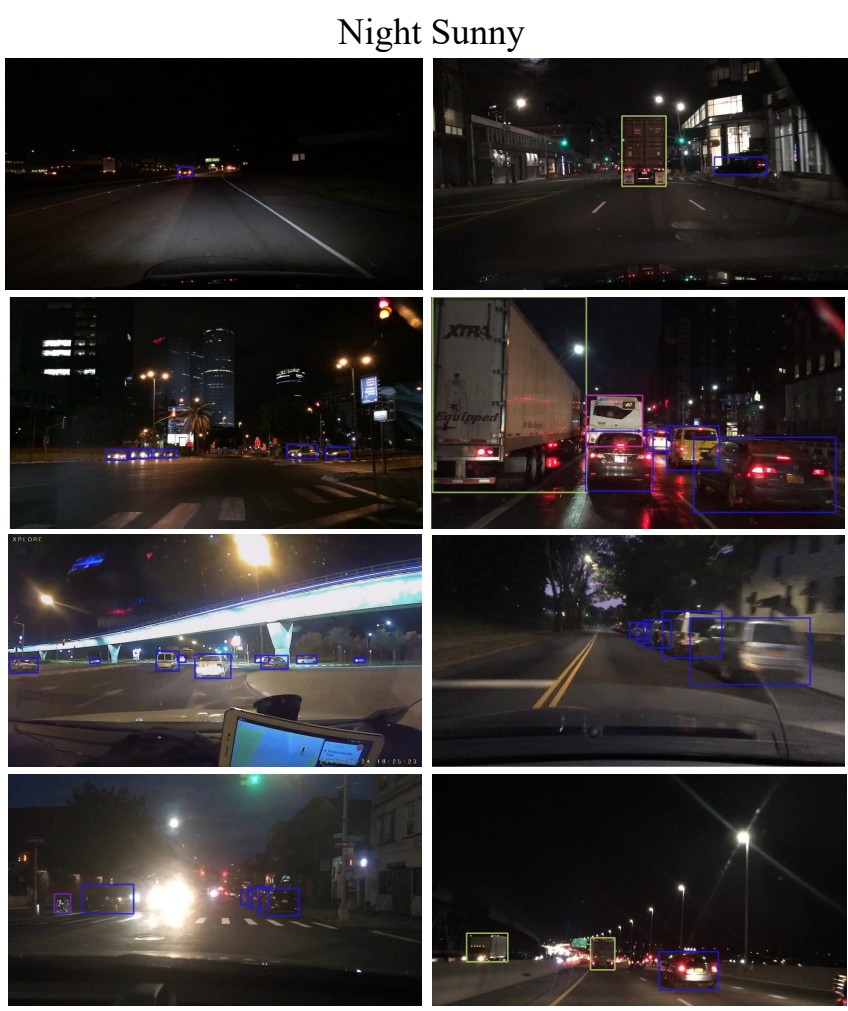

Figure 11: Visualization results of object detection on Daytime Sunny→Night Sunny.

Clipart

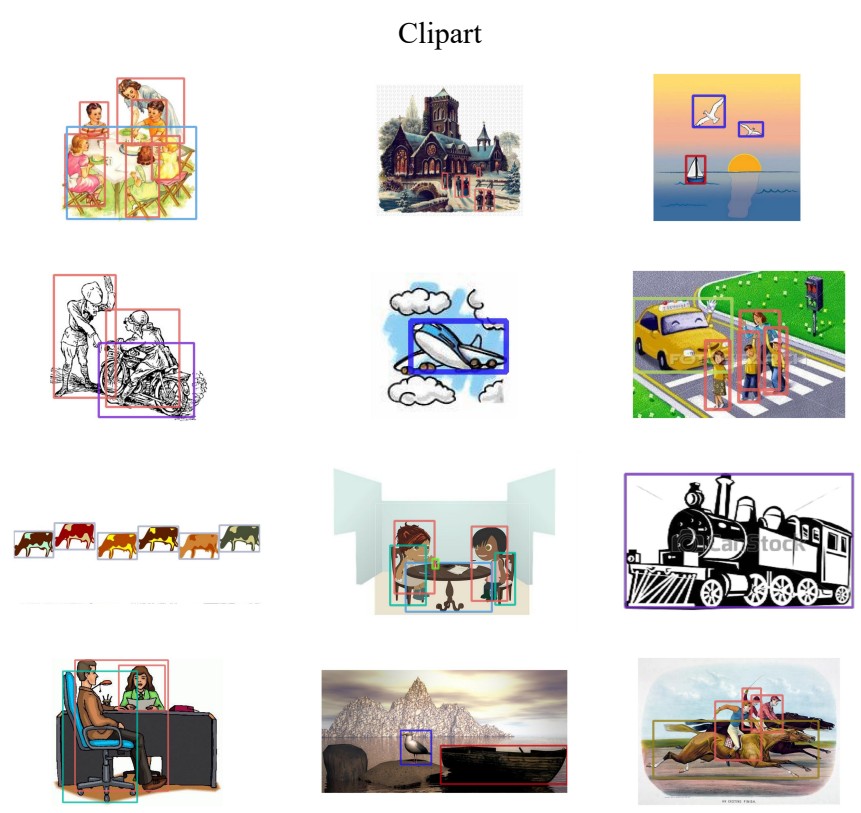

Figure 12: Visualization results of object detection on PASCAL VOC 2007→Clipart.

WaterColor

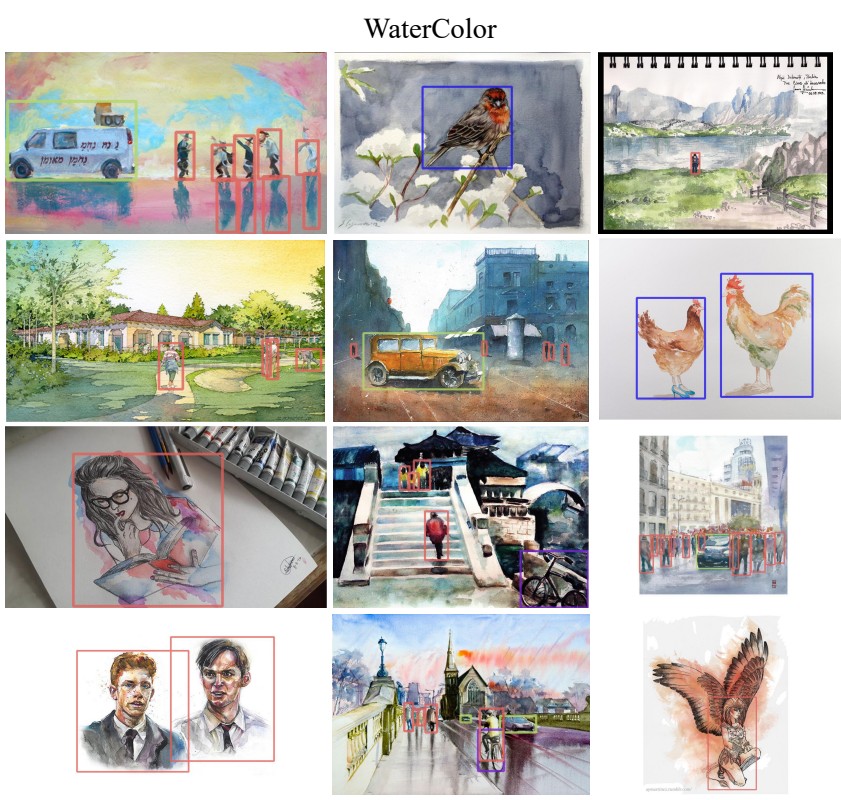

Figure 13: Visualization results of object detection on PASCAL VOC 2007→WaterColor.

