# OpenReview forum: "Phrase Grounding-based Style Transfer for Single-Domain Generalized Object Detection"
_ICLR.cc/2024/Conference — ICLR 2024 Conference Withdrawn Submission_

### Official Review · Reviewer_G99H · 2023-10-31

**Soundness:** 3 good
**Presentation:** 3 good
**Contribution:** 3 good
**Rating:** 6
**Confidence:** 4

**Summary:**

The paper proposes a phrase grounding-based style transfer (PGST) approach for single-domain generalized object detection. The authors leverage the grounded language-image pre-training model (GLIP) to learn object-level, language-aware, and semantic-rich visual representations. They define textual prompts for each target domain and use them to train the PGST module, which performs style transfer from the source domain to the target domain. The authors evaluate their approach on five different weather driving benchmarks and achieve significant improvements over existing methods.

**Strengths:**

- The paper addresses an important and challenging problem of single-domain generalized object detection.
- The proposed PGST approach is novel and leverages the strengths of the GLIP model.
- The evaluation results show significant improvements over existing methods on diverse weather driving benchmarks.

**Weaknesses:**

- The experimental evaluation could benefit from more detailed analysis and discussion of the results.
- The paper could provide more insights into the reasons behind the observed improvements

**Questions:**

1. Can the authors provide more insights into the limitations of the proposed approach and potential directions for future research?
2. How sensitive is the performance of the proposed approach to the choice of textual prompts? Have the authors experimented with different prompt designs and evaluated their impact on the results?

---

> ### Author Response · Authors · 2023-11-18
> **Response to Reviewer G99H**
>
> Thanks for your diligent review work and valuable comments on our paper. In response to your feedback, our detailed replies are as follows:
>
> **Weakness 1: More detailed experimental analysis and discussions.**
>
> Thank you for the valuable suggestion you provided.
>
> In responding to the **Reviewer JhCL**'s comments on **Weakness 2**, we conducted experimental analysis on the catastrophic forgetting issue in large-scale vision-language models and verified that the proposed PGST module can alleviate this problem to some extent.
>
> Addressing the **Reviewer JhCL**'s comments on **Weakness 3**, we experimentally analyzed the performance degradation when using a general prompt set. This is attributed to the difficulty for the model's style transfer module to adaptively filter out irrelevant prompt phrases based on the target domain.
>
> Regarding the **Reviewer JhCL**'s comments on **Weakness 4**, we experimentally analyzed how weather-unrelated prompts decrease the model's performance.
>
> In response to the **Reviewer Ci7j**'s comments on **Weakness 1**, we experimentally analyzed the effectiveness of the designed module to verify that the significant improvement is not solely derived from GLIP.
>
> Addressing the **Reviewer Ci7j**'s comments on **Weakness 2**, we similarly conducted experimental analysis on the catastrophic forgetting issue in large-scale vision-language models.
>
> Regarding the **Reviewer Ci7j**'s comments on **Question 2**, we experimentally analyzed the impact of different fine-tuning strategies on the model.
>
> **Weakness 2: More insights behind the observed improvements.**
>
> Thank you for the valuable suggestion you provided.
>
> As per your suggestions, we will include more insights to the reasons behind the observed improvements in our revised submission.
>
> As explained in the response to **Reviewer Ci7j**'s comments on **Question 2**, we hold the view that the effectiveness of large-scale model-based domain generalization approaches derives from exposure to relevant visual concepts. For example, a large-scale model might not have encountered images of driving at night, but it may have learned the essential visual concept of nighttime. Hence, prompts can be employed to evoke these visual concepts in large-scale models, effectively addressing the distribution gap between the source and target domains. Specifically, by leveraging prompts to induce implicit visual concepts in large models, the proposed style transfer module PGST combines these visual concepts to generate the stylistic characteristics of the unseen target domain. Subsequently, it applies this stylistic transformation to the visual features of the source domain, ultimately enhancing the generalization performance of the model trained on the source domain to the unseen target domain.
>
> **Question 1: More insights into limitations and potential future directions.**
>
> Thanks for this kind advice. We will include more discussions of limitations and future directions in our revised submission.
>
> Firstly, our model heavily relies on the availability of well-defined and representative textual prompts for each target domain. If these prompts are not adequately defined or do not accurately represent the characteristics of the target domain, the performance of our model may be impacted. Future research could explore methods for automatically generating or adapting textual prompts to improve the model' s adaptability to unseen target domains.
>
> While the adopted AdAIN has achieved state-of-the-art performance, there is still room for improvement in terms of the style transfer strategy. For example, incorporating a local feature loss [1] could enhance the quality of local visual transfer, thereby better serving our object-level transfer tasks.
>
> [1] AdaAttN: Revisit Attention Mechanism in Arbitrary Neural Style Transfer, Liu et al. ICCV'2021.
>
> **Question 2: Choice of textual prompts.**
>
> Thanks for this kind comment. In our response to the **Reviewer JhCL**, we have conducted experimental analyses on the choice of textual prompts. Firstly, we investigated the effect of a general set of prompts containing various weather conditions on model performance (**Reviewer JhCL Weakness 4**). Subsequently, we experimentally analyzed the impact of weather-unrelated prompts on model performance (**Reviewer JhCL Weakness 3**). The experimental results indicate that choosing either of these prompts would decrease the model's performance. This suggests the importance of designing prompts that are as relevant as possible to each target domain.
>
> We hope your concerns will be resolved and the rating of the paper can be increased accordingly. Thank you!

---

> > ### Comment · Reviewer_G99H · 2023-11-22
> >
> > I am pleased to see that you have added experimental analysis on the catastrophic forgetting issue, and that you mentioned the limitation of your model requiring high quality for representative textual prompts. This has explained my doubts, and I will maintain my rating.

---

> ### Author Response · Authors · 2023-11-22
> **Further response to Reviewer G99H**
>
> Thank you for your thoughtful review and positive feedback.
>
> We would like to bring your attention to our last response to **Reviewer JhCL**, where we specifically addressed the limitation of our model in relation to the need for high-quality and representative textual prompts.
>
> In the latest submission, we have also developed a set of templates for general prompts that are independent of domain-specific textual prompts. As show in following table, the results of the two prompt designs are nearly identical. **This indicates the robustness of our proposed approach to prompt design, which does not necessitate high-quality and representative textual prompts.**
>
> | Method                           | Night Sunny | Dusk Rainy    | Night Rainy   | Daytime Foggy |
> |----------------------------------|:-------------:|:---------------:|:---------------:|:---------------:|
> | PGST  (w/ domain-specific prompt)                             | 46.6 | **45.1**| 27.2| **42.7**|
> | PGST (w/ general prompt) | **47.9**        | 44.5          | **28.4**          | 42.5          |
>
> We have updated the experimental results with the general prompt as the **default setting**, while presenting the results using the previous domain-specific prompt as a reference or comparison. Furthermore, all these modifications have been highlighted in **blue** in the revised submission.
>
> Once again, we sincerely appreciate your feedback and the opportunity to address your concerns. We recognize the significance of prompt design in our work and have taken your concerns seriously. We have made substantial efforts to tackle this limitation and have provided additional insights in our responses. We hope that our explanations and clarifications have effectively addressed your concerns regarding prompt design.

---

> > ### Author Response · Authors · 2023-11-23
> > **Further discussion with Reviewer G99H**
> >
> > Dear Reviewer,
> >
> > As the deadline is approaching rapidly, we kindly ask for your prompt feedback on our revised manuscript and the additional information provided in our rebuttal. Your evaluation and suggestions are highly valuable to us, and we are eager to know if our efforts have successfully addressed your concerns.
> >
> > Additionally, we believe that the modifications made and the additional insights provided have significantly enhanced the quality and impact of our research. We kindly request your thoughtful evaluation, taking into account the improvements we have implemented. We sincerely hope for your consideration in potentially revising the rating or score assigned to our paper.
> >
> > If you have any other requests or require further assistance, please feel free to let us know. We are dedicated to addressing any remaining questions or concerns you may have.
> >
> > Thank you once again for your valuable contribution to our work. We eagerly await your feedback and the possibility of an improved rating.
> >
> > Best regards,
> >
> > Authors.

---

### Official Review · Reviewer_Ci7j · 2023-11-01

**Soundness:** 2 fair
**Presentation:** 3 good
**Contribution:** 2 fair
**Rating:** 6
**Confidence:** 3

**Summary:**

This paper presents Phrase Grounding-based Style Transfer (PGST) for single-domain generalized object detection. PGST aligns image regions with textual prompts, enabling the model to perform well in multiple unseen domains. It outperforms existing methods and achieves large improvement across various benchmarks.

**Strengths:**

This paper is the first work to apply GLIP model to single-domain generalized object detection. In terms of novelty and performance improvement, it is a success. However, I'm a little concerned about the fair comparison, please see Weaknesses.

**Weaknesses:**

- When comparing with other SOTA methods, the comparison is not fair. The proposed method is based on GLIP, while previous methods are based on Faster R-CNN. Apparently, GLIP has much stronger capacity than Faster R-CNN. It's hard to say how much improvement comes from the proposed design, instead of GLIP network architecture or pre-trained data.
- Will fine-tuning GLIP with PGST degenerate the GLIP's original performance, like its performance on COCO, Flicker30k entities?

**Questions:**

I assume the following questions are open problems and not considered as the weaknesses of this paper:
- Since GLIP has been pre-trained on so many data, there might be data leakage of target domain data. If so, does this really follow the problem setting of "single-domain generalization" ?
- For the source domain augmentation with prompt, this paper uses a full-model tuning strategy. Have the authors tried prompt/linear probing and what the performance is ?

---

> ### Author Response · Authors · 2023-11-18
> **Response to Reviewer Ci7j (I)**
>
> Thanks for your diligent review work and valuable comments on our paper. In response to your feedback, our detailed replies are as follows:
>
> **Weakness 1: Comparison fairness.**
>
> Thanks for this valuable comment.
>
> GLIP has its own object detector and we are the first to apply GLIP to single domain generalized object detection problem. Moreover, we adopt the region-phrase alignment loss in the object detector to optimize the style transfer module. Therefore, it is non-trivial to achieve a fair comparison you mentioned. To emphasize the crucial role of the proposed design in performance improvements, we conducted experiments and recorded the results in the following table, including Faster-RCNN, C-Gap (based on the large-scale pre-trained model CLIP and employing Faster-RCNN), GLIP, and our proposed design. It can be observed that, on one hand, our proposed method exhibits a significant improvement over GLIP. On the other hand, the relative improvement of our proposed design over GLIP is more substantial compared to the relative improvement of C-Gap over Faster-RCNN. These experimental results indicate the effectiveness of our proposed design. Additionally, in the response to **Weakness 2 of Reviewer JhCL**, the proposed design could effectively mitigate catastrophic forgetting problem in GLIP.
>
> | Method      | Night Sunny         | Dusk Rainy            | Night Rainy           | Daytime Foggy         |
> |-----------|:-------------------:|:---------------------:|:---------------------:|:---------------------:|
> | Faster RCNN | 33.5                | 26.6                  | 14.5                  | 31.9                  |
> | C-Gap       | 36.9(10.1\%)        | 32.3(21.4\%)          | 18.7(29.0\%)          | 38.5(20.7\%)          |
> | GLIP        | 32.2                | 32.1                  | 16.1                  | 32.1                  |
> | Ours        | **40.0(24.2\%)** | **39.4(22.7\%)**| **22.0(36.6\%)** | **39.8(24.0\%)** |
>
> **Weakness 2: Fine-tuning GLIP with PGST degenerates GLIP's original performance.**
>
>
> Thanks for this valuable comment.
>
> What you mentioned is the common challenge of catastrophic forgetting in large-scale models. When the model undergoes fine-tuning on an interested dataset, it may forget its capabilities on other datasets due to overfitting issues, especially when the fine-tuning dataset follows different distribution from the test dataset. As shown in the following table, the performance of the GLIP model fine-tuned on VOC decreased by 10.5\% when evaluated on COCO.
>
> When we fine-tuned the model using the VOC dataset and the proposed style transfer module PGST for Clipart dataset, we observed an improvement from 36.1\% to 36.6\%. Therefore, our proposed PGST does not exacerbate the issue of catastrophic forgetting. The fundamental reason lies in the overfitting of the VOC dataset and the distribution difference between the VOC and COCO datasets.
>
> It is worth noting that the reason why our GLIP (Fine-tuned on VOC+PGST) performs worse than GLIP on the COCO dataset is mainly because our style transfer module PGST is designed for the Clipart dataset. Since COCO is a comprehensive dataset with diverse domains, the generalization from VOC to COCO involves a compound domain generalized object detection task, posing significant challenges in designing prompts, which is beyond the scope of this submission. However, it is an intriguing research direction that we may explore in the future.
>
> | Model                           | mAP of COCO |
> |-------------------------------|:-----------:|
> | GLIP                            | 46.6        |
> | GLIP (Fine-tuned on VOC)        | 36.1        |
> | GLIP (Fine-tuned on VOC + PGST) | 36.6        |

---

> ### Author Response · Authors · 2023-11-18
> **Response to Reviewer Ci7j (II)**
>
> **Question 1: Data leakage problem of unseen target domain.**
>
> Thanks for this valuable comment.
>
> As you pointed out, when using GLIP or other large-scale vision-language models, the potential issue of data leakage may exist. However, both previous research [1] and the findings in this paper indicate that fine-tuning a large-scale model on a specific domain can result in a decrease in its generalization performance on another different domain. The main reasons leading to the poor generalization issue are both catastrophic forgetting and distribution shift issues, as discussed in **Weakness 2**.
>
> Moreover, we believe that the success of the large-scale model based domain generalization approaches does not solely stem from encountering specific data from the target domain but rather from exposure to relevant visual concepts. For instance, a large-scale model may not have seen driving images at night, but it may have encountered the crucial visual concept of nighttime. Therefore, we can utilize prompts to evoke these visual concepts in large-scale models, thereby bridging the distribution gap between the source and target domains.
>
> [1] Investigating the Catastrophic Forgetting in Multimodal Large Language Models, Zhai et al., arxiv 2023.
>
> **Question 2: Prompt/linear probing.**
>
> Thanks for this kind advice. Firstly, we apologize for any unclear expression in the submission. In fact, we analyzed the impact of fine-tuning strategies in *Table 11* of the revised submission. Additionally, as shown in the following table, we tested the performance of two fine-tuning strategies: full tuning and prompt tuning, and drew the following conclusions: 1) The full tuning strategy outperforms the prompt tuning strategy; 2) The introduction of the proposed style transfer module PGST under the prompt tuning strategy results in larger improvements compared to the one under the full tuning strategy.
>
> | Method           | Mode          | Night Sunny | Dusk Rainy | Night Rainy | Day Foggy |
> |----------------|-------------|:-----------:|:----------:|:-----------:|:---------:|
> | Src. Aug.        | Full tuning   | 43.8        | 40.1       | 24.0        | 37.4      |
> | Src. Aug. + PGST | Full tuning   | 47.9        | 44.5       | 28.4        | 42.5      |
> | Src. Aug.        | Prompt tuning | 32.2        | 32.1       | 16.1        | 32.1      |
> | Src. Aug. + PGST | Prompt tuning | 40.0        | 39.4       | 22.0        | 39.8      |
>
> We hope your concerns will be resolved and the rating of the paper can be increased accordingly. Thank you!

---

> > ### Comment · Reviewer_Ci7j · 2023-11-23
> >
> > Thanks for authors' detailed response and added experimental results. I will increase my rating to 6. Good luck.

---

> ### Author Response · Authors · 2023-11-23
> **Author-reviewer discussion**
>
> Dear reviewers,
>
> We genuinely value your important feedback. As the deadline for the author-reviewer discussion phase is approaching, we would like to ascertain if you have any other lingering concerns regarding our revised submission.
>
> We sincerely appreciate your commitment and dedication in evaluating our submission. Please feel free to inform us if you require any further clarification or have additional suggestions.
>
> Best Regards,
>
> Authors.

---

### Official Review · Reviewer_JhCL · 2023-11-01

**Soundness:** 3 good
**Presentation:** 3 good
**Contribution:** 2 fair
**Rating:** 6
**Confidence:** 5

**Summary:**

This paper tackles single-domain generalization tasks for object detection. In this work, authors leverage the GLIP model to estimate different unseen target domains via their style transfer module, PGST, and text prompts which describe the object categories in the new domains. Once the different styles are learned both image and text encoders are finetuned to achieve the best performance. The state-of-the-art results are shown for the standard benchmarks.

**Strengths:**

1. The manuscript is well-written and easy to follow
2. Experiments are shown on standard domain generalization and adaptation benchmarks.
3. Though the method takes inspiration from C-Gap(2023) in terms of using text prompts for domain generalization, using GLIP instead of CLIP seems to be a more reasonable direction for object detection tasks. The strong improvement over C-Gap and other baselines goes to show that.

**Weaknesses:**

1. The novelty of this work is using their PGST and GLIP for domain generalization tasks. However, a previous work PODA[1,2], which is not cited in this paper, implements a module similar to PGST using text prompts. This reduces the novelty of the current work. The authors should discuss this in the paper and propose what makes their PGST different from PODA.

2. This work's performance is still much better than in PODA, so there is some merit. But if we remove PGST from the contribution (because of similarity w.r.t PODA ), is the contribution just integrating GLIP for domain generalization?

3. All prompts used in this work directly correspond to the test domains. Why not have a general set of prompts showing all possible weather descriptions? How does that affect the performance? For example: a quick ChatGPT prompts for different weather scenarios and time of the day.
```
1. "an image taken on a rainy day during the morning."
2. "an image taken on a cloudy day during the evening."
3. "an image taken on a snowy day during the night."
4. "an image taken on a sunny day during the early morning."
5. "an image taken on a foggy day during the late afternoon."
6. "an image taken on a stormy day during the twilight."
7. "an image taken on a clear day during the midnight."
8. "an image taken on a windy day during the golden hour."
9. "an image taken on a partly cloudy day during the dusk."
10. "an image taken on a misty day during the early evening."
```

4. Also, what if the prompts are unrelated to the weather , does it degrade the performance? These studies will be useful in judging sensitivity to the prompt's choice and design.

5. It is not clear how the best model is chosen. Please refer to Gulrajani et Lopez-Paz , In search of lost domain generalization , ICLR'21 to indicate what strategy was used. This is crucial for the reproducibility of the method.

[1] PODA: Prompt-driven Zero-shot Domain Adaptation, Fahes et. al. ICCV'23

[2] PØDA: Prompt-driven Zero-shot Domain Adaptation, Fahes et. al. arxiv, 2022

**Questions:**

Please have a look at the weakness for my major concerns. Based on the answers, I am willing to change my rating.

---

> ### Author Response · Authors · 2023-11-18
> **Response to Reviewer JhCL (I)**
>
> Thanks for your diligent review work and valuable comments on our paper. In response to your feedback, our detailed replies are as follows:
>
> **Weaknesses 1: Differences between PGST and PODA.**
>
> Thanks for this valuable comment.
>
> Fahes et al. propose PODA to address the domain generalized semantic segmentation task. Specifically, PODA leverages a pre-trained contrastive vision-language model (CLIP) to optimize affine transformations of source features, steering them towards the target text embedding while preserving their content and semantics. To achieve this, PODA utilizes prompt-driven feature augmentation inspired from adaptive instance normalization (AdaIN). Then, they show that these prompt-driven augmentations can be used to deal with the domain generalized semantic segmentation task. The main differences between PODA and the PGST proposed in this submission can be summarized in the following four aspects:
>
> [1] PODA: Prompt-driven Zero-shot Domain Adaptation, Fahes et al. ICCV'23.
>
> 1. **Task**. PODA aims to deal with the semantic segmentation task, while the proposed PGST is specifically designed for the object detection task. Object detection task is more complex and challenging compared to semantic segmentation task, as it involves both classification and regression simultaneously.
> 2. **Model**. PGST adopts GLIP instead of CLIP in PODA. As you pointed out, using GLIP instead of CLIP seems to be a more reasonable direction for the object detection task.
> 3. **Optimization Strategy for Style Transfer**. PODA aligns the visual feature of a whole source image with the target text embedding. To achieve this, PODA minimizes the cosine distance between the source visual features and target text features in CLIP latent space, to optimize the style transfer module. In contrast, the proposed PGST aligns the object-level visual feature of a source image with the target phrase embedding. To optimize the style transfer module, we employ the region-phrase alignment loss in GLIP which could enhance object-level style transfer while preserving the content and semantics.
> 4. **Prompt Design**. As PODA mainly focuses on image-level style transfer, they use prompt like "an image taken on a rainy day during the morning" to specify the "rainy-morning" domain. In contrast, as the proposed PGST aims to realize object-level style transfer, we adopt prompt like "a photo of a person taken on a rainy day during the morning, a photo of a car taken on a rainy day during the morning..." to specify objects within the "rainy-morning" domain.
>
> Notably, both the PGST and PODA are inspired from the classic style transfer algorithm AdAIN as it can effectively manipulate the style information with a small set of parameters. In future work, we will explore more mature style transfer algorithms to enhance the performance of our proposed model, such as [2] and [3].
>
> [2] Learning Dynamic Style Kernels for Artistic Style Transfer, Xu et al. CVPR'23.
>
> [3] U-GAT-IT: Unsupervised Generative Attentional Networks with Adaptive Layer-Instance Normalization for Image-to-Image Translation, Kim et al. ICLR'20.
>
> As per your suggestion, we will include these discussions on the differences between PODA and PGST in the revised submission.
>
> **Weaknesses 2: Contributions of our study.**
>
> Thanks for this valuable comment.
>
> We have discussed the four distinctions between our proposed model and PODA in **Weakness 1**. We hope you will reconsider our contributions. Additionally, we would like to draw your attention to the following contribution.
>
> The proposed PGST module can alleviate the catastrophic forgetting issue in the GLIP model. This issue arises due to overfitting during the fine-tuning process of large-scale models. The following table shows the forgetting issue of GLIP on VOC and Clipart datasets. It could be observed that GLIP model exhibits test performance of 0.64 and 0.33 on VOC and Clipart datasets (zero-shot), respectively. However, when separately fine-tuned on VOC and Clipart, although GLIP can achieve good results on the corresponding fine-tuning datasets, their performance on the other dataset tends to degrade compared to zero-shot setting. In contrast, with the proposed PGST module, the forgetting issue on VOC and Clipart is significantly alleviated, improving from 0.54 to 0.66 and 0.23 to 0.36.
>
> | Model | mAP of VOC | mAP of Clipart |
> | --- |:----------:|:--------------:|
> | GLIP (Zero-shot)                           | 0.64       | 0.33           |
> | GLIP (Fine-tuned on VOC)                   | 0.71       | 0.23           |
> | GLIP (Fine-tuned on Clipart)               | 0.54       | 0.50           |
> | **GLIP (Fine-tuned on Clipart + PGST)** | **0.66**       | 0.50           |
> | **GLIP (Fine-tuned on VOC + PGST)**   | 0.71       | **0.36**           |

---

> ### Author Response · Authors · 2023-11-18
> **Response to Reviewer JhCL (II)**
>
> **Weakness 3: A general set of prompts.**
>
> Thanks for this valuable comment.
>
> As you highlighted, it is crucial to evaluate the model's performance with a general set of prompts encompassing all possible weather descriptions. In our study, we employ specific-domain prompt tailored to the test domains, aiming to enhance the model's representations for each domain and improve its performance in specific scenarios. However, if we adopt a general set of prompts, the challenge arises in adaptively filtering out irrelevant weather descriptions to achieve precise style transfer in unseen target domains. This direction adds an intriguing dimension to our research. To address your concern, we create a general set of prompts for each category by incorporating various weather and time-related words to assess their impact on generalization performance. For instance, in the "car" category, we include prefix phrases such as "daytime, dusk, and night," as well as suffix phrases like "in sunny, rainy, and foggy scenes." The following table showcases the model's performance with the general set of prompts, revealing a potential decline in performance. We will include these analysis experiments in our revised submission.
>
> | Method                           | Night Sunny | Dusk Rainy    | Night Rainy   | Daytime Foggy |
> |----------------------------------|:-------------:|:---------------:|:---------------:|:---------------:|
> | PGST                             | **46.6** | **45.1**| **27.2**| **42.7**|
> | PGST (w/ general set of prompts) | 44.9        | 43.5          | 26.1          | 41.8          |
>
> **Weakness 4: Unrelated prompts**.
>
> Thanks for this valuable comment.
>
> As per your suggestion, we have conducted analysis experiments with prompts that are unrelated to weather shown in the following table. As you pointed out, the performance will be degraded as these prompts make it difficult to learn the essential differences between domains and the effectiveness of style transfer is consequently diminished. To produce prompts that are unrelated to weather, we utilize ChatGPT to generate prefixes and suffixes for each category of the target domain. In the table, **A** represents adding the prefix "dreamlike setting" and the suffix "in the scene exudes a boring atmosphere"; **B** represents adding the prefix "ineffective" and the suffix "crawl slowly through the desert". We will include these analysis experiments in our revised submission.
>
> | Method                                | Night Sunny | Dusk Rainy    | Night Rainy   | Daytime Foggy |
> |-------------------------------------|:-----------:|:-------------:|:-------------:|:-------------:|
> | PGST                                  | **46.6** | **45.1** | **27.2** | **42.7** |
> | PGST (w/ unrelated prompt **A**) | 44.0        | 40.5          | 24.2          | 37.8          |
> | PGST (w/ unrelated prompt **B**) | 43.2        | 40.7          | 24.6          | 37.2          |
>
> **Weakness 5: Model selection strategy.**
>
> Thanks for this kind advice.
>
> Gulrajani and Lopez-Paz [4] argue that domain generalization algorithms without a model selection strategy should be regarded as incomplete, where the model selection includes choosing hyper-parameters, training checkpoints, architecture variants). Moreover, they introduce three model selection strategy, i.e., training-domain validation set, Leave-one-domain-out cross-validation, test-domain validation set (oracle). Our work mainly focuses on single domain generalized object detection task, which is a more challenging task. Therefore, we adopt the third model selection strategy, i.e., test-domain validation set (oracle). In order to make our experimental results more convincing, we will submit the code for reproducibility. We will cite [4] and clarify the model selection strategy adopted in our work, and include them in section of implementation details of our revised submission.
>
> [4] Gulrajani and Lopez-Paz, In Search of Lost Domain Generalization, ICLR'21.
>
> If you think these responses address your concerns, please consider increasing your score. Thank you!

---

> ### Author Response · Authors · 2023-11-21
> **Author-reviewer discussion**
>
> Dear reviewers,
>
> We sincerely appreciate your valuable feedback.
>
> As the deadline for the author-reviewer discussion phase is approaching, we would like to check if you have any other remaining concerns about our paper.
>
> We sincerely thank you for your dedication and effort in evaluating our submission. Please do not hesitate to let us know if you need any clarification or have additional suggestions.
>
> Best Regards,
>
> Authors.

---

> ### Comment · Reviewer_JhCL · 2023-11-21
> **The approach is more domain adaptation than generalization.**
>
> Thank you authors for your replies.
> My major concerns now are based on Weaknesses 3 and 5.
>
>
> W3: The performance degrades when the set of prompts is general(even when they represent a shift due to weather change). This means one has to be aware of the exact target domains and this is against the domain generalization spirit that the target domains are unknown. The proposed methodology is closer to domain adaptation (as the target domain is known) than generalization.
>
> W5: The model selection strategy used in this work is test-domain validation setting, which is different from C-Gap, where they use training-domain validation. Hence, the metrics computed are not directly comparable with C-Gap. The test-domain validation is the weakest form compared to others as one has access to the test domain during training.
>
> Based on the replies, it seems more like domain adaptation i.e. without access to target images but still being aware of the exact target domain and validation on the same.

---

> ### Author Response · Authors · 2023-11-22
> **Further response to Reviewer JhCL**
>
> Thank you to the reviewers for the positive feedback on our previous responses during the discussion and for raising additional concerns about some of our responses.
>
> Further response for **W5**:
>
> Sorry, I apologize sincerely. We made a mistake in describing the model selection strategy here. **We have confirmed that we indeed employ the same model selection strategy as C-Gap, namely, the training domain validation strategy.**
>
> Our experimental details are as follows: During the model training process, without accessing any data or images from the target domain, we divided a single source domain (Daytime-Sunny) into a training subset and a validation subset. Specifically, the training subset consists of 19,395 images, and the validation subset involves the remaining 8,313 images. **Then, we select the model that is trained on the training subset and could achieve the highest mAP on the validation subset before being tested on the unseen target domain.** In order to provide more convincing evidence, we have included our open-source code in the supplementary materials. Additionally, in the revised manuscript, we have corrected our description of the model selection which are highlighted in **blue** color (Section of Implementation Details).
>
> Once again, we apologize for our mistake, and we hope that the above response addresses your concerns regarding our submission.
>
> Furher response for **W3**:
>
> Thank you for the concerns raised by the reviewer regarding our submission. Indeed, C-Gap [5] is a signifcant work and we find many aspects of their experimental setup worth following, including the model selection strategy you mentioned and the design of the general prompt.
>
> [5] Vidit et al, CLIP the Gap: A Single Domain Generalization Approach for Object Detection, CVPR'23.
>
> The extensive additional experiments that needed to be conducted in our previous response. In order to quickly address your comments, we used only a portion of the source domain data for data augmentation and ran a relatively small number of iterations. Unfortunately, this led to less satisfactory results in our previous response, and we apologize for this. Over the past few days, **we have gradually completed the full experimental process, strictly following the settings in the experimental details of our submission**.
>
> The obtained results, as shown in the table below, indicate that after incorporating the general prompt you mentioned, our approach performs better on domains with significant differences (e.g., Night Sunny, Night Rainy) compared to the domain-specific prompt. However, on domains with smaller differences (e.g., Dust Rainy, Daytime Foggy), the performance is slightly worse with the general prompt. We speculate that domains with smaller differences are more susceptible to interference from the prompts for the domain with significant differences in the general prompt. **However, the results of the two are almost identical, indicating that our proposed approach is quite robust to prompt design, which is also the aspect you were most concerned about. Furthermore, even with the use of the general prompt, our method still exhibits superiority compared to C-Gap.**
>
> | Method                           | Night Sunny | Dusk Rainy    | Night Rainy   | Daytime Foggy |
> |----------------------------------|:-------------:|:---------------:|:---------------:|:---------------:|
> | C-Gap | 36.9       | 32.3         | 18.7        | 38.5         |
> | PGST  (w/ domain-specific prompt)                             | 46.6 | **45.1**| 27.2| **42.7**|
> | PGST (w/ general prompt) | **47.9**        | 44.5          | **28.4**          | 42.5          |
>
> **In summary, we will use the general prompt you mentioned to ensure that we are in the domain generalization setting and to maintain fairness in comparison with C-Gap.** We updated the experimental results with the general prompt as the **default setting** in our submission, and presented the results with the domain-specific prompt as a reference or comparison, demonstrating the robustness of our proposed approach to prompt design. Moreover, these modifications have all been highlighted in **blue** in the revised submission.
>
> To further convince you of our results, we have submitted our code and documentation in the supplementary material, and update our results in the revised version of our submission. We apologize for the hastily obtained experimental results in our previous response, and hope that the updated results and analysis will address your concerns about our submission.
>
> Once again, thank you for your suggestions on our prompt design. Your review has a positive impact on our work, allowing us to refine our proposed framework. **Such a general prompt design is meaningful in clearly defining that the submission addresses domain generalization problem rather than domain adaptation problem, and it contributes to improving the robustness of the model to the prompt design.**

---

> > ### Author Response · Authors · 2023-11-23
> > **Author-reviewer discussion**
> >
> > Dear reviewers,
> >
> > We genuinely value your insightful feedback.
> >
> > With the deadline for the author-reviewer discussion stage drawing near, we would like to inquire if there are any lingering concerns about our revised submission.
> >
> > We wholeheartedly appreciate your commitment and diligence in evaluating our submission. Please feel free to reach out if you require further clarification or have additional suggestions.
> >
> > Best Regards,
> >
> > Authors.

---

> ### Author Response · Authors · 2023-11-23
> **Author-reviewer discussion**
>
> Dear Reviewer,
>
> As the rebuttal period is coming to an end, and I have not received further feedback from you regarding our previous responses. Considering the limited time remaining, it is crucial for us to address any concerns you may have and strive to resolve them to the best of our abilities.
>
> We highly value your feedback and suggestions, and we genuinely want to meet your expectations. If there is any additional information you need, further discussion you would like to have, or any other issues you would like us to address, please let us know.
>
> We would appreciate the opportunity to have a final discussion with you to ensure that our responses are satisfactory to you. We sincerely appreciate your time and patience and look forward to receiving your feedback as soon as possible.
>
> If there are any other requests or if you require further assistance, please feel free to let us know.
>
> Best regards,
>
> Authors.

---

### Author Response · Authors · 2023-11-18
**General Response**

We first thank for all reviewers for pointing out the strength of our work:

1. The manuscript is well-written and easy to follow (\#Reviewer JhCL).

2. Strong improvements over C-Gap and other baselines (\#Reviewers JhCL, Ci7j, and G99H).

3. The proposed approach is the first work to apply GLIP model to single-domain generalized object detection (\#Reviewer Ci7j), which is an important and challenging problem (\#Reviewer G99H). Moreover, using GLIP instead of CLIP seems to be a more reasonable direction for object detection tasks (\#Reviewers JhCL).

In order to make our experimental results more convincing, we will submit the code for reproducibility. Additionally, we have revised our submission, and the modified areas are highlighted in blue in the updated submission. Finally, we hope the following individual response could resolve the concerns and answer the questions raised by the reviewers.